# Electromagnetic fields alter the motility of metastatic breast cancer cells

Ayush Arpit Garg [1], Travis H. Jones[1], Sarah M. Moss[2], Sanjay Mishra[3,4], Kirti Kaul[3,4], Dinesh K. Ahirwar[3,4], Jessica Ferree[1], Prabhat Kumar[1], Deepa Subramaniam[5], Ramesh K. Ganju[3,4], Vish V. Subramaniam[1,4] & Jonathan W. Song [1,4]

Interactions between cells and their environment influence key physiologic processes such as their propensity to migrate. However, directed migration controlled by extrinsically applied electrical signals is poorly understood. Using a novel microfluidic platform, we found that metastatic breast cancer cells sense and respond to the net direction of weak (~100 μV cm$^{-1}$), asymmetric, non-contact induced Electric Fields (iEFs). iEFs inhibited EGFR (Epidermal Growth Factor Receptor) activation, prevented formation of actin-rich filopodia, and hindered the motility of EGF-treated breast cancer cells. The directional effects of iEFs were nullified by inhibition of Akt phosphorylation. Moreover, iEFs in combination with Akt inhibitor reduced EGF-promoted motility below the level of untreated controls. These results represent a step towards isolating the coupling mechanism between cell motility and iEFs, provide valuable insights into how iEFs target multiple diverging cancer cell signaling mechanisms, and demonstrate that electrical signals are a fundamental regulator of cancer cell migration.

---

[1] Department of Mechanical and Aerospace Engineering, The Ohio State University, Columbus, OH 43210, USA. [2] Department of Biomedical Engineering, The Ohio State University, Columbus, OH 43210, USA. [3] Department of Pathology, College of Medicine, The Ohio State University, Columbus, OH 43210, USA. [4] Comprehensive Cancer Center, The Ohio State University, Columbus, OH 43210, USA. [5] College of Medicine, The Ohio State University, Columbus, OH 43210, USA. Correspondence and requests for materials should be addressed to V.V.S. (email: subramaniam.1@osu.edu) or to J.W.S. (email: song.1069@osu.edu)

Directional cell migration is fundamental to multiple physiological processes, such as wound healing, embryonic morphogenesis, and immune cell trafficking[1]. Moreover, it is when malignant cells in a tumor acquire the capacity to migrate that cancer transforms from a local and largely curable condition to a metastatic, systemic, and deadly disease[2]. Important contributors to metastasis include biomolecular gradients of growth factors and chemokines from the primary tumor microenvironment, which impart potent migratory cues that help initiate and maintain tumor cell migration[3]. For example, epidermal growth factor (EGF) modulates the motility machinery of EGF receptor (EGFR) expressing tumor cells that includes polarized signaling, cell adhesion, and cytoskeletal remodeling[4]. In addition, the accumulation of oncogenic mutations over time in metastatic cancer cells promotes their transition to a migratory phenotype[5].

While cell migration due to extrinsic chemokines and accumulated genetic mutations has been widely studied, a better understanding of the role of physical interactions, and their interplay with biochemical changes, may provide significant insights into tumor metastasis and the foundation for new non-invasive therapeutic approaches[6]. The existence of endogenous electric fields (EFs) and electric currents, and their biological responses in non-excitable cells have been documented in physiological settings for several centuries[7]. A common approach for applying EFs to cells in vitro is with the contact-based electrodes. These direct current EFs (dcEFs), which result in current flow, have been shown to stimulate migration and provide directional guidance cues to both normal and tumor cells[8]. This phenomenon is interchangeably referred to as electrotaxis or galvanotaxis[8,9], and multiple mechanisms, involving PI3K/Akt signaling, ion channels, or receptor polarization, have been proposed to govern electrotactic responses to dcEFs[10]. It is noteworthy that only strong dcEFs (>0.5 V cm$^{-1}$) are reported to induce directional migration while low strength dcEFs (<0.5 V cm$^{-1}$) have no effect[7]. Although strong dcEFs can steer or reorient migrating cells, to our knowledge they have not been demonstrated to hinder or slow down migration driven by extrinsic drivers of motility such as chemokine gradients.

Another approach for electrically treating cells is with alternating EFs, which are generated in accordance with Faraday's Law of electromagnetic induction[11]. Since these fields are inherently non-contact, they are devoid of electrochemical reactions associated with contact electrodes used to produce dcEFs. Alternating EFs also differ from dcEFs in that they lack bulk electric current flow in the medium surrounding cells. Recently, another class of alternating symmetric (in time) low-frequency EFs (≥1 V cm$^{-1}$ at 100–300 kHz) known as tumor treating fields (TTFs) have shown promising clinical outcomes for glioblastoma patients[12]. TTFs have also been shown to hinder migration and invasion of glioma cells and glioma-initiating cells[13]. TTFs have been reported to primarily target dividing cells and arrest cell proliferation, and though their governing mechanism has been studied extensively[14–16], the mechanisms that alter the migratory behavior of metastatic cancer cells remain unknown[13]. It is important to recognize that even though inductively generated EFs (or iEFs) alternate in direction, it is nevertheless possible to create a time-averaged net directional field effect with temporally asymmetric magnetic field excitation. Such asymmetrically alternating iEFs applied at intensities (~1 μV cm$^{-1}$) orders of magnitudes lower than the intensities of dcEFs (≥0.5 V cm$^{-1}$) required to induce electrotaxis prominently hindered the migration of highly metastatic breast cancer cells in vitro in a direction-dependent manner at 100 kHz[11]. However, the underlying governing mechanism controlling this inhibitory response is unknown.

The present study seeks to elucidate the mechanisms by which metastatic breast cancer cells sense and respond to low frequency, weak (<100 μV cm$^{-1}$), and asymmetrically alternating iEFs. Using a novel microfluidic bi-directional microtrack (MBDM) assay, we observed changes in the characteristics of spontaneously migrating (i.e., in the absence of a chemotactic gradient) MDA-MB-231 and MCF10CA1a breast cancer cells[17–19], and compared them with normal MCF10A cells under iEF treatment. These cell lines were chosen in order to understand the effects of iEFs on metastatic breast cancer cells and benchmark it against its effects on normal epithelial breast cells (MCF10A), and the MCF10CA1a enabled understanding the lineage specific effects of iEFs as this cell line is a cancerous version derived from normal MCF10A cells[19]. We confirmed that the Akt pathway plays a vital role for cells to sense the direction of the applied iEFs and in modulating their migration responses. We found that iEFs downregulated EGFR activation and also prevented formation of actin-rich filopodia in breast cancer cells in the presence of EGF. However, the signal transduction pathway from EGFR to actin was not necessarily conserved, but was dependent on the genetic background of the cell. Further, we observed selective hindering of EGF-promoted cancer metastasis through synergistic treatment with iEFs and MK2206, a potent pan-Akt inhibitor. Therefore, our experimental results in the absence or presence of chemotactic gradients, demonstrate the ability of migrating breast cancer cells to not only sense the presence of iEFs but also sense their net direction. These results underscore iEFs as potent controllers of cell migration, the importance of genetic background of different cell lines, and the role of biochemical signals in influencing how cells sense and process these responses by electrotransduction.

## Results

**iEFs alter metastatic breast cancer cell motility.** Temporally asymmetric iEFs were produced with a custom Helmholtz coil (Fig. 1a–c and Supplementary Fig. 1). We developed a custom MBDM assay (Fig. 1d–f) that enabled real-time monitoring of cell motility dynamics. The MBDM assay with parallel microtrack arrays (~20 μm width and height), replicate the topography of preexisting paths formed by vessels, extracellular matrix fibers, and white matter tracts in the brain that guide migrating cancer cells in vivo[20]. Various cancer cells (including those of the breast) have exhibited spontaneous and persistent migration in microtracks of comparable dimensions in vitro[21]. The unique design of the MBDM assay sustained stable chemokine gradients over 12 h (Supplementary Fig. 2). Directed migration was quantified by cell speed and the dimensionless quantity, persistence (Supplementary Fig. 3). The latter is defined as the capacity to maintain on average, a single direction of motion[22–24]. A high persistence (near 1) indicates a cell's ability to maintain a singular direction of migration while low persistence (near 0) indicates frequent directional changes or lack of net migration alltogether[24].

Cells were seeded in the center port of the MBDM assay where they can bi-directionally migrate into opposing collection chambers (Fig. 1d). Cell movement in response to iEFs applied primarily in the direction of migration (parallel) or against the direction of migration (antiparallel) was compared within the same microfluidic device. The field direction is that with the higher peak magnitude, and higher time-averaged field strength (Supplementary Fig. 1).

As expected, MDA-MB-231 (Supplementary Movie 1) and MCF10CA1a cells, spontaneously (i.e., EGF(−)) migrated in the absence of chemokine or inhibitor at the median of the mean speeds of 0.28 μm min$^{-1}$ and 0.41 μm min$^{-1}$, respectively (Fig. 2a, c). iEF treatment alone, applied parallel to the direction

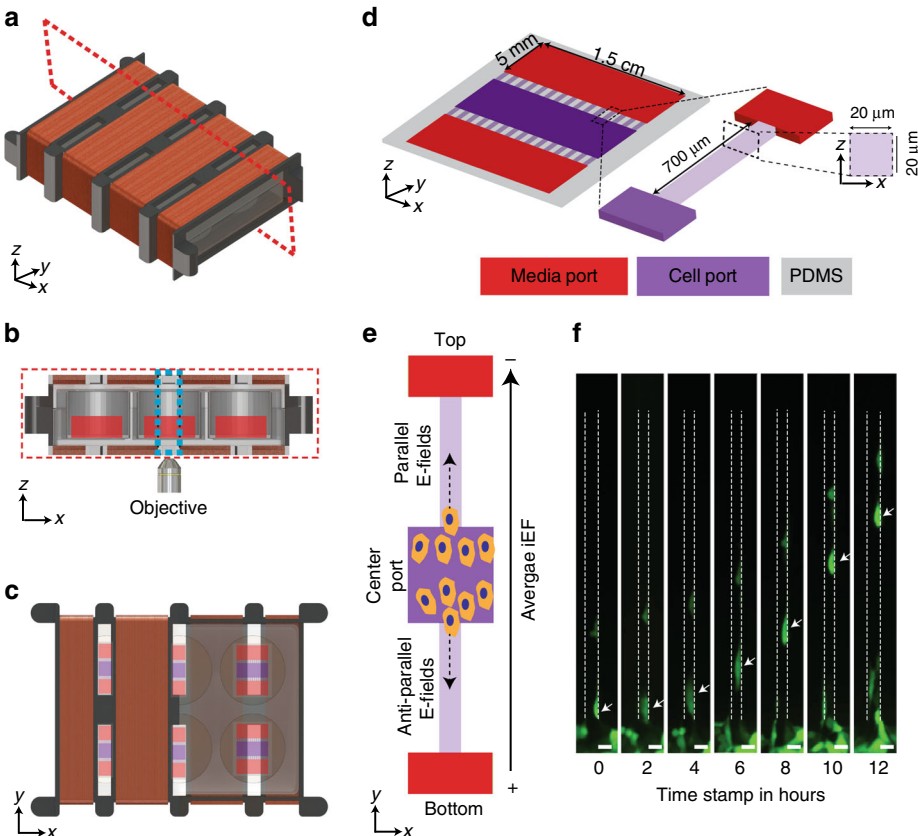

**Fig. 1** Experimental setup for quantifying cell migration in response to iEF treatment. **a** Isometric view of the Helmholtz coil used to apply inductive electric field (iEF) treatment on migrating cells. **b** Cross-section cut (plane marked with dotted red line in **a** indicating the location of the microfluidic bi-directional migration (MBDM) assay) and its relative position with the microscope objective. **c** Top view of the Helmholtz coil showing the location of devices in **d** and the microscope observation window. **d** Schematic of MBDM assay. Cells are seeded in the center port (purple) and are tracked as they migrate to the outer ports (red) through the microtracks (inset) connecting them. **e** Cells from the center port can migrate into the opposing microtracks and migrate either toward the top or bottom media ports and under the influence of iEFs applied either parallel or antiparallel to the direction of cell migration. **f** Time-lapse images of GFP-tagged MDA-MB-231 cell migrating through a single microtrack. (*Scale Bar = 20 μm)

of migration had no effect on migration speeds (Fig. 2a, c). In contrast, standalone iEF treatment in the antiparallel direction resulted in median of the mean migration speeds increasing by 45% for MDA-MB-231 and 25% for MCF10CA1a, compared with untreated controls (Fig. 2a, c).

Established external regulators of persistence include chemotactic factors and mechanical cues (e.g., topography of the extracellular matrix[3]), but the role of EFs in modulating this migration response is not well understood. For MDA-MB-231 cells, standalone antiparallel iEFs significantly increased persistence compared with untreated controls (Fig. 2b). Significant increase in persistence of MCF10CA1a cells was observed with iEFs (parallel and antiparallel) (Fig. 2d). We also used the modified transwell migration assay[11] to determine the effects of standalone iEF treatment on number of cells migrated. iEF treatment alone had no significant (p = 1.000) effect (neither stimulatory nor inhibitory) on the total number of MDA-MB-231 and MCF10CA1a cells that spontaneously (i.e., EGF(−)) migrated across the Transwell membrane (~10 μm thick) compared with untreated controls (Supplementary Fig. 4).

In summary, both breast cancer cell lines (MDA-MB-231 and MCF10CA1a) exhibited a directional response with EGF(−)/iEF (+) treatment by migrating faster and with greater persistence with iEFs applied in the antiparallel direction when compared with untreated controls. The response of iEFs on normal MCF10A cells was also quantified. iEFs did not induce any migratory response in MCF10A cells, which remained in their

monolayer structure maintaining their non-migratory phenotype. Therefore, unlike dcEF based galvanotaxis[25], iEFs (parallel or antiparallel) did not induce migration of MCF10A cells, but only altered speeds and persistence of cancer cells with a migratory phenotype.

**iEFs potently hinder EGF-induced breast cancer cell motility.** Next, we assessed the role of iEFs in modulating cell motility promoted by the pleiotropic signaling molecule EGF. Physiologically, breast cancer metastasis is promoted by biomolecular EGF gradients[26]. Furthermore, its cognate receptor EGFR is overexpressed in breast cancer cells and is correlated with poor prognosis[27]. With stable EGF gradients (Supplementary Fig. 2), median mean migration speeds of MDA-MB-231 (Supplementary Movie 1) and MCF10CA1a cells increased significantly (p < 0.001) compared with untreated controls (Fig. 3a). Interestingly, when parallel iEFs were applied in the direction of EGF-gradient promoted motility, MDA-MB-231 cells median of the mean migration speeds decreased (Supplementary Movie 1) by 20% and returned to untreated control speeds (0.31 μm min$^{-1}$). In contrast, antiparallel iEFs had no observable effect on EGF-gradient promoted migration. Therefore, mean speeds of MDA-MB-231 cells migrating with EGF(+) exhibited a directional response to iEFs. MDA-MB-231 cells (Fig. 3b) migrated with an average persistence of 0.84 for EGF(+), which was not statistically different (p = 1.000) compared with the untreated controls. iEF

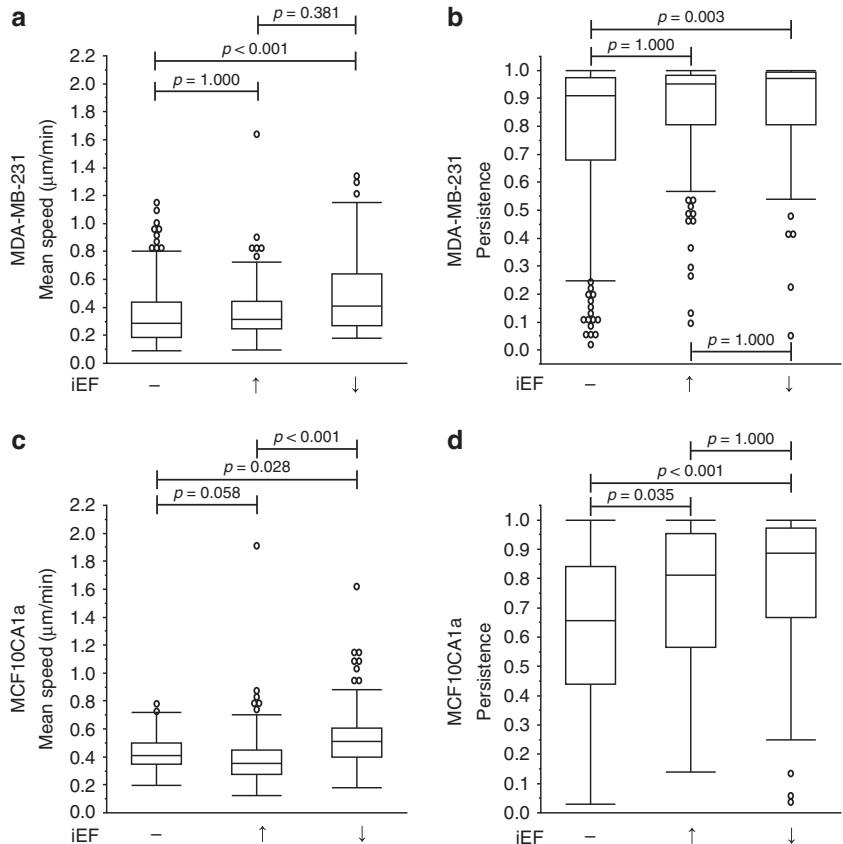

**Fig. 2** Antiparallel iEF treatment increases the migration speeds of breast cancer cells migrating without exogenous EGF gradients. **a** MDA-MB-231: iEFs applied antiparallel to the direction of migration increased migration speeds compared with untreated controls but had no effects when applied parallel to the direction of migration. **b** Antiparallel iEFs increased cell persistence. **c** MCF10CA1a: antiparallel iEFs increased migration speeds. **d** iEFs bi-directionally increased cell persistence. The normal MCF10A cells do not migrate under these conditions, and iEFs have no effect on the migratory behavior of these cells under these set of conditions. All data is presented as box plots show the minimum, 1st quartile, median, 3rd quartile, and maximum. All data pooled from three independent biological replicates for each condition. (Nonparametric independent samples Kruskal–Wallis test)

treatment (parallel or antiparallel) had no observable effect on the persistence of these cells. For MCF10CA1a, iEFs applied in either direction decreased median of the mean migration speeds in the presence of EGF gradients and nullified the stimulatory effects of EGF on their migration speeds (Fig. 3c), returning them to untreated control levels. Surprisingly, iEF treatment increased persistence of MCF10CA1a cells bi-directionally (Fig. 3d). Therefore, iEFs reduced the overall migratory potential of MDA-MB-231 cells, while partially hindering the migratory potential of MCF10CA1a cells. The MCF10A cells also migrate under EGF gradients (Fig. 3e, f). Interestingly, antiparallel iEFs increased mean migration speeds of MCF10A but had no effect with parallel iEFs. However, parallel iEFs increased persistence of MCF10A for EGF(+), but were unchanged with antiparallel iEFs.

We also used the modified Transwell migration assay to assess the number of EGF-stimulated cells migrated with and without iEF treatment[11]. As expected, EGF(+)/iEF(−) stimulation markedly promoted the transmigration of the MDA-MB-231 cells compared with untreated controls (Supplementary Fig. 4). Interestingly, iEF treatment potently hindered EGF-promoted migration of MDA-MB-231 for both iEF directions. However, iEFs had no observable effects on migration numbers for MCF10CA1a. Unlike the effects on speed and persistence, iEFs potently hindered EGF-promoted migration of MCF10A cells in the modified Transwell assay. Collectively, results from both migration assays (MBDM and modified Transwell) clearly demonstrate that iEFs selectively hinder the EGF-promoted

motility of breast cancer cells. The ability to sense and respond to direction of applied iEFs varied between cell lines, indicating that response to iEFs is also dependent on cell lineage.

**iEF treatment downregulates EGFR activation.** To determine the direct effects of iEF on EGFR activation and to explore possible mechanisms controlling their effect on cell motility, we examined the spatial distribution (immunofluorescence staining), activation, and expression (western blotting) of EGFR. One of the sites, Tyr-1068, is involved in critical signaling pathways triggered by its phosphorylation[28]. Further, receptor clustering or aggregation has been shown to result in EGFR activation[29,30].

EGFR was uniformly distributed in untreated controls for MDA-MB-231 cells (Fig. 4a, Supplementary Fig. 5A). As expected, negligible phosphorylation of EGFR was observed at the Tyr-1068 site for controls (Fig. 5a and Supplementary Fig. 6). Upon standalone iEF treatment, we observed striking changes in EGFR spatial distribution through formation of aggregates and clusters (Fig. 4a, Supplementary Fig. 5A). However, EGFR clustering did not result in autophosphorylation and EGFR remained in its inactive state (Fig. 5a and Supplementary Fig. 6). Similarly, treatment of the MDA-MB-231 cells with only EGF, also resulted in EGFR clustering (Fig. 4a, Supplementary Fig. 5A), but to a lesser degree than with iEF treatment alone. As expected, EGF(+)/iEF(−) treatment resulted in EGFR activation with clear increase in phosphorylation (Fig. 5a and

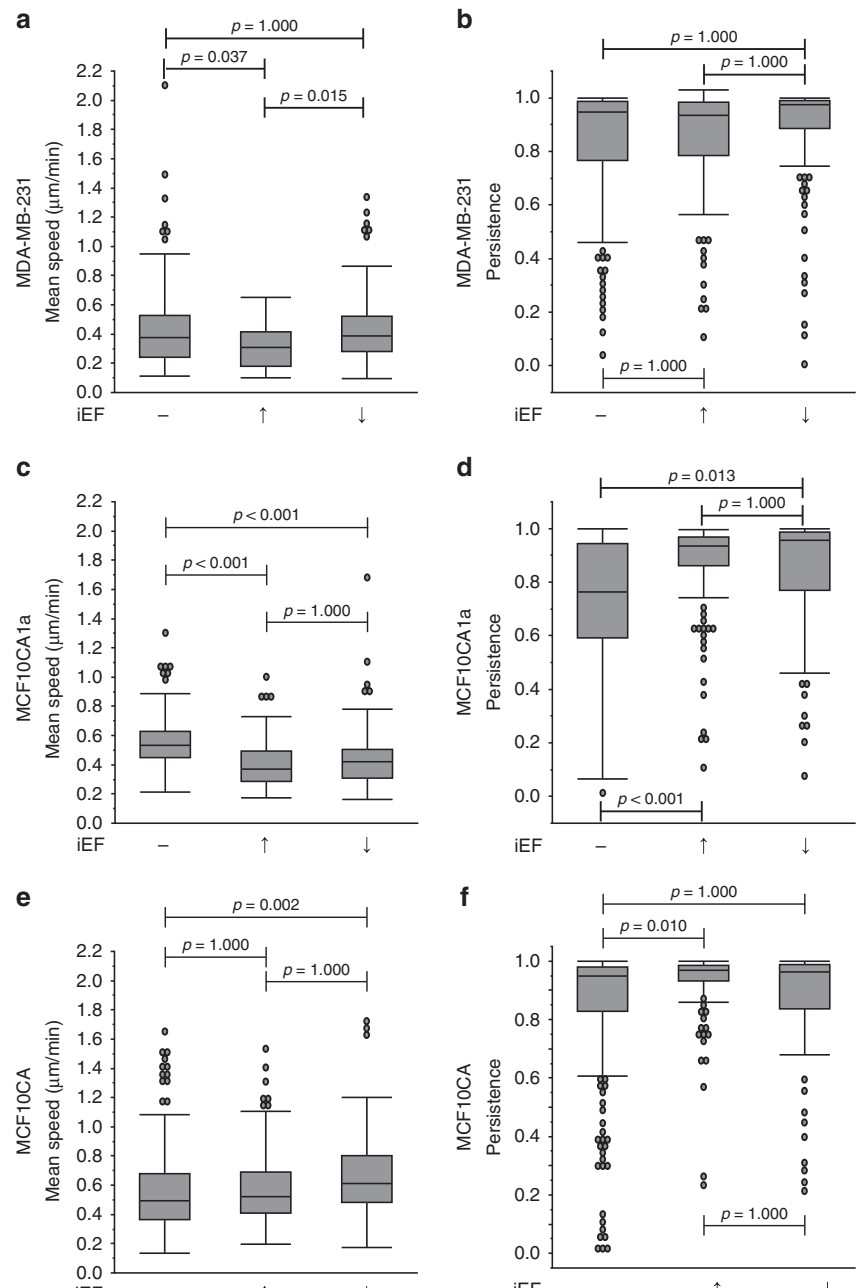

**Fig. 3** iEF treatment decreases the motility of breast cancer cells migrating under EGF gradients. **a** MDA-MB-231: parallel iEFs decreased migration speeds by 21% compared with cells migrating under EGF gradients, but no inhibitory effects were observed with antiparallel iEFs. **b** iEFs had no effect (parallel or antiparallel) on persistence of cells migrating under EGF gradients. **c** MCF10CA1a: iEFs bi-directionally inhibit EGF-gradient promoted motility. **d** iEFs bi-directionally increase persistence of these cells migrating under EGF-gradients. **e** MCF10A: antiparallel iEFs increased migrating speeds of cells migrating under EGF-gradients. **f** Parallel iEFs increased persistence of these cells migrating under EGF-gradients. All data is presented as box plots show the minimum, 1st quartile, median, 3rd quartile, and maximum. All data pooled from three independent biological replicates for each condition. (Nonparametric independent samples Kruskal–Wallis Test)

Supplementary Fig. 6). The clustering observed with EGF and EGFR activation is consistent with previous reports[30,31]. Furthermore, iEF(+)/EGF(+) treatment did not alter the EGFR clustering previously observed with iEF treatment alone; EGFR continued to display clustered states (Fig. 4a, Supplementary Fig. 5A). However, iEF treatment on EGF-treated MDA-MB-231 cells downregulated EGFR phosphorylation by ~21% (Fig. 5b and Supplementary Fig. 6). Therefore, EGF-induced activation of EGFR was downregulated with iEF(+) despite continued receptor clustering. There were no changes in total EGFR (t-EGFR)

expression for any of the above conditions (Fig. 5c). Consistent with these results, the ratio of phosphorylated EGFR (p-EGFR) to t-EGFR (total EGFR) followed the same trend as p-EGFR, where iEF(+)/EGF(+) treatment on MDA-MB-231 cells downregulated p-EGFR/t-EGFR ratios by ~24%, compared with iEF(−)/EGF(+) (Fig. 5d and Supplementary Fig. 6). Therefore, iEFs hinder EGF-promoted motility of MDA-MB-231 cells by downregulating EGFR phosphorylation.

No discernible differences in the EGFR spatial distribution were observed with MCF10CA1a cells with iEF(+), as were

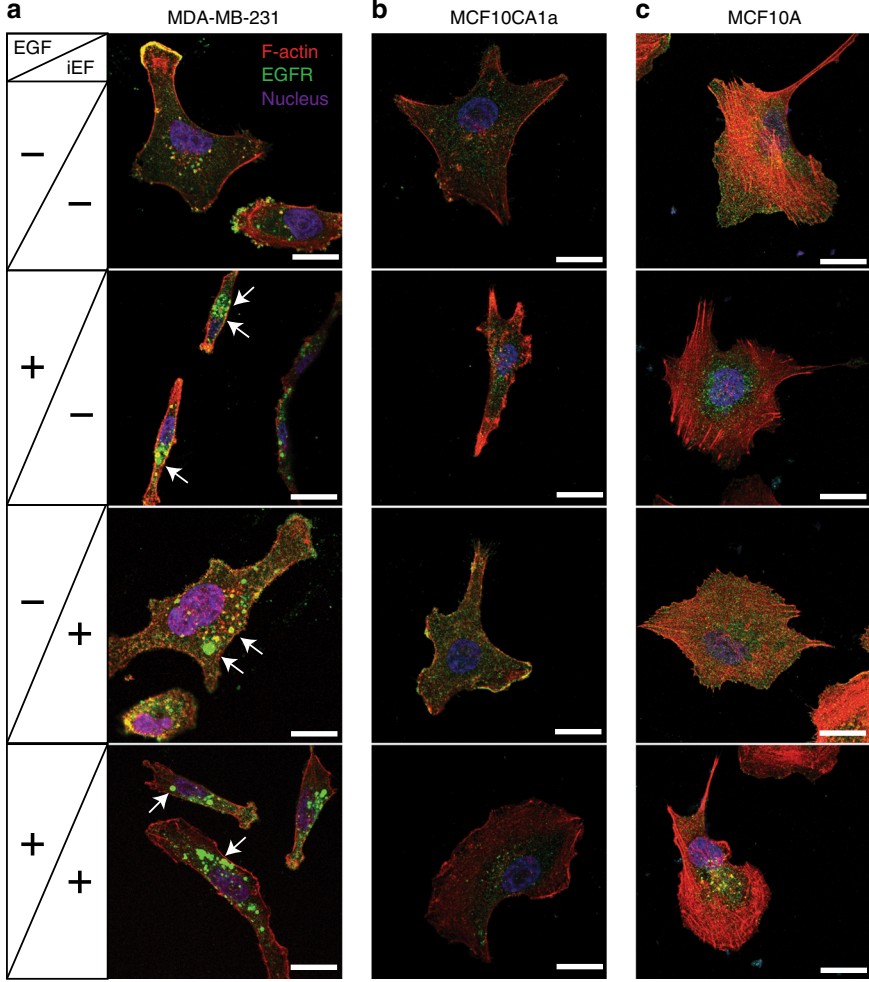

**Fig. 4** iEF treatment promotes EGFR aggregation in MDA-MB-231 breast cancer cells. **a** MDA-MB-231: iEFs induce EGFR clustering and cause receptor aggregation independent of EGF treatment. **b** MCF10CA1a: iEFs have no effect of EGFR distribution, however, iEF treatment in presence of EGF results in downregulation of EGFR expression. **c** MCF10A: iEFs have no effect on EGFR aggregation/clustering. (Red—Actin, Blue—nucleus, and Green—EGFR, scale bar is 10 μm). The standalone split channel EGFR images shown in Supplementary Fig. 5

observed with MDA-MB-231 cells (Fig. 4b, Supplementary Figs. 5B). Striking though was that iEF(+)/EGF(+) down-regulated EGFR phosphorylation by ~39% (Fig. 5e, f and Supplementary Fig. 6) and downregulated t-EGFR expression by ~50% (Fig. 5g and Supplementary Fig. 6) when compared with iEF(−)/EGF(+) treatment. Since t-EGFR expression was down-regulated for this particular cell line, it was not surprising that receptor activation decreased. Therefore, p-EGFR/t-EGFR ratio remained unaffected by iEF(+)/EGF(+) treatment for MCF10CA1a cells (Fig. 5h and Supplementary Fig. 6). These results again imply that genetic variation within cell lines makes them susceptible to iEF treatments differently.

EGFR phosphorylation in MCF10A cells remained completely unaffected by iEFs. iEFs caused no changes in EGFR spatial distribution (Fig. 4c, Supplementary Fig. 5C). EGFR activation (Fig. 5i, j and Supplementary Fig. 6) and expression (Fig. 5k, l and Supplementary Fig. 6) were unchanged with iEF treatment for these cells. Clearly, iEF treatment in the absence of EGF gradients had no effect on migration or protein activation in MCF10A cells.

**iEFs alter intracellular F-actin distribution**. Cytoskeletal F-actin is a critical component of the cell directional response machinery.

Its polymerization is very important in membrane extension (i.e., lamellipodia/pseudopodia/filopodia), formation of cell-substrate attachments, and contractile force and traction[31]. Distribution of F-Actin for cells migrating along the microtracks of the MBDM assay is quantified by a nondimensional quantity called the polarization ratio (PR), ranging from 0 to 1 (see Image Acquisition and Processing). A PR of 0 indicates no preferential F-actin aggregation at the leading and/or trailing edges of the cells, while a PR of 1 indicates preference for F-actin localization in migrating cells.

For migrating MDA-MB-231 cells (Fig. 6a, b), the center of the PR distribution for EGF(−)/iEF(−) was 0.34. For parallel iEFs and EGF(−), PR distribution centers decreased to 0.15. However, this was not a significant change compared with controls, and cell migration speeds were also comparable to controls. Antiparallel iEFs and EGF(−) increased the PR distribution center to 0.45, but this was also not statistically significant. Nevertheless, this increase mirrored the higher migration speeds measured for this case compared with controls. The PR distribution center for MDA-MB-231 migrating with EGF(+)/iEF(−) changed to 0.62, indicating a higher proportion of cells with F-actin localization at the leading/trailing edges compared with controls (i.e., EGF (−)/iEF(−)). With EGF gradients, parallel iEFs decreased the

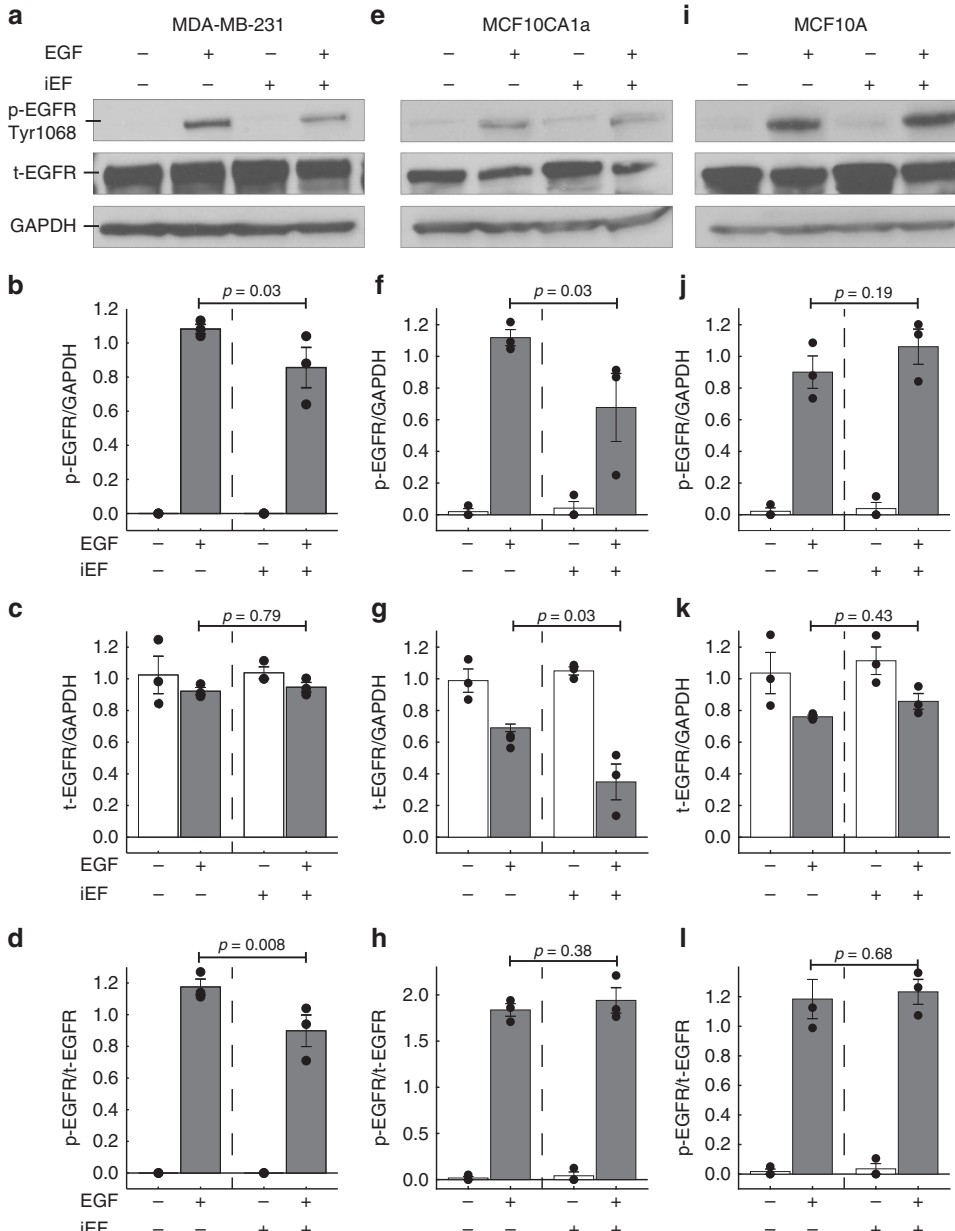

**Fig. 5** iEF treatment downregulates EGFR phosphorylation in breast cancer cells. **a** MDA-MB-231: western blot analysis shows that iEFs downregulate EGFR phosphorylation in EGF-treated cells. **b** Densitometry analysis for phosphorylated EGFR (p-EGFR) levels. **c** Densitometry analysis for total EGFR (t-EGFR) levels. **d** Ratio of p-EGFR to t-EGFR levels. **e** MCF10CA1a: Western blot analysis shows that iEFs downregulate EGFR phosphorylation and expression in EGF-treated cells. **f** Densitometry analysis for phosphorylated EGFR (p-EGFR) levels. **g** Densitometry analysis for total EGFR (t-EGFR) levels. **h** Ratio of p-EGFR to t-EGFR levels. **i** MCF10A: western blot analysis shows that iEFs have no effect on EGFR phosphorylation or expression. **j** Densitometry analysis for phosphorylated EGFR (p-EGFR) levels. **k** Densitometry analysis for total EGFR (t-EGFR) levels. **l** Ratio of p-EGFR to t-EGFR levels. All data presented as mean ± SEM. (Unpaired two-tailed Student $t$ test, $*p < 0.05$, all data were pooled from three independent biological replicates for each condition)

PR center by ∼55% to 0.28 compared with EGF(+)/iEF(−). In contrast, antiparallel iEFs showed no changes in the PR center.

For MCF10CA1a (Fig. 6c, d), the PR center increased with iEF (+)/EGF(−) (independent of iEF direction) in the MBDM assay compared with untreated controls. In contrast, actin-rich filopodia formation was reduced (downward shift in PR distribution center) for iEF(+)/EGF(+). iEF(+) (independent of the iEF direction) nullified the pro-migratory effects of EGF-gradients for this metastatic cell line. Therefore, the increased actin-rich filopodia formation for iEF(+)/EGF(−) versus the opposite observed for iEF(+)/EGF(+), follows the trends in migration speeds observed for this cell line. Moreover, the pro-

migratory response for EGF(−)/iEF(+) and anti-migratory response for EGF(+)/iEF(+) clearly indicate the presence of two different signaling axes.

EGF-gradients induced migration of MCF10A cells, as expected (Fig. 6e, f). Actin-rich filopodia observed along EGF gradients for cells in the microtracks of the MBDM assay confirm their growth factor stimulated migratory response. iEF treatment, parallel or antiparallel, had no effect on the actin distribution of these cells. The PR center remained unchanged for EGF(+)/iEF (+) when compared with EGF(+)/iEF(−).

In summary, actin distributions for iEF(+) show trends consistent with decreased mean migration speeds of MDA-MB-

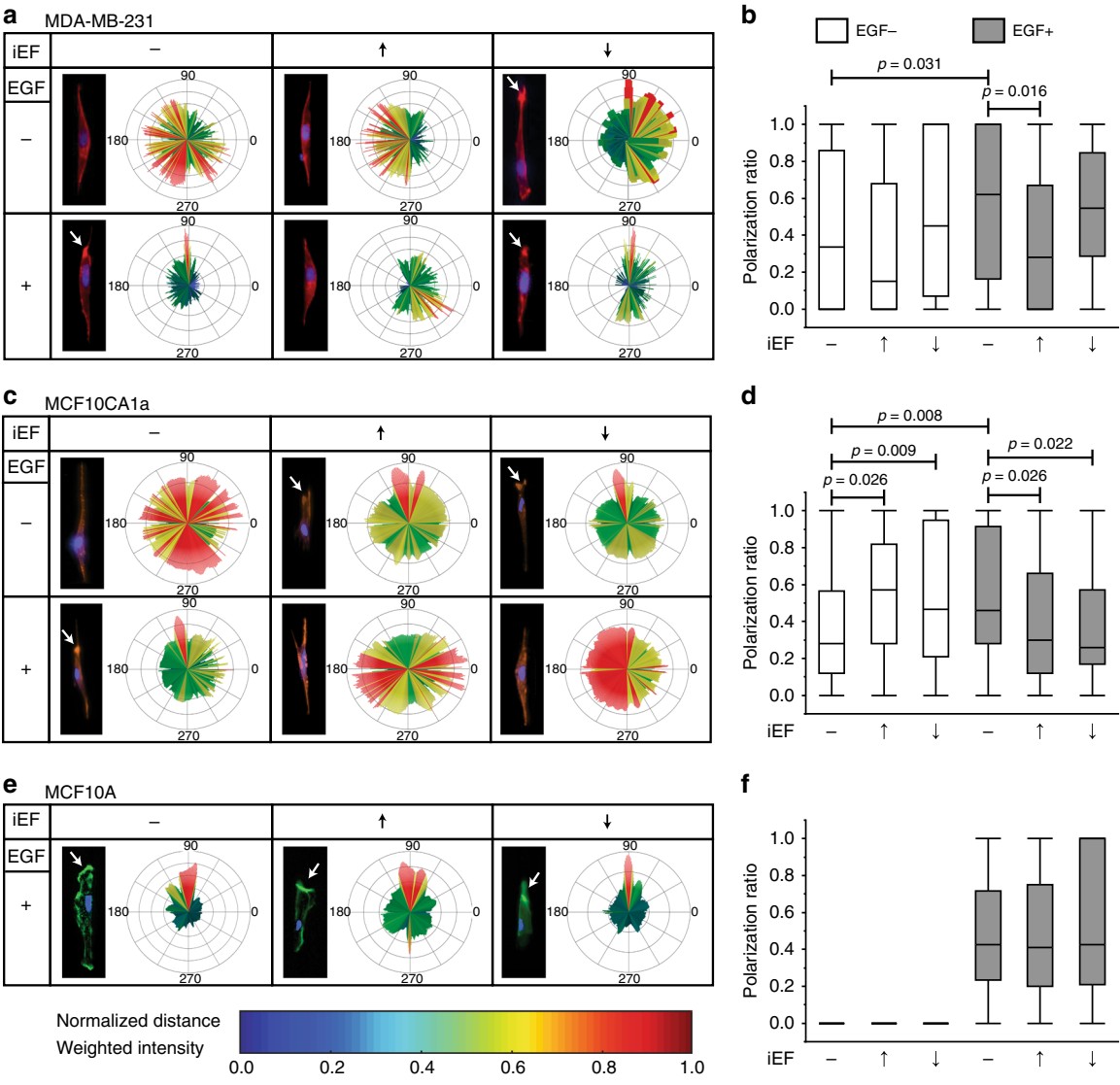

**Fig. 6** iEF treatment inhibits EGF-promoted actin aggregation at the leading edge of migrating cancer cells. **a** MDA-MB-231: immunofluorescence images of MDA-MB-231 cells stained for F-actin with phalloidin (red) and nuclei with DAPI (blue). **b** Quantification of the F-actin polarization ratio. **c** MCF10CA1a: immunofluorescence images of MCF10CA1a cells stained for F-actin with phalloidin (red) and nuclei with DAPI (blue). **d** Quantification of the F-actin polarization ratio. **e** MCF10A: immunofluorescence images of MCF10CA1a cells stained for F-actin with phalloidin (red) and nuclei with DAPI (blue). **f** Quantification of the F-actin polarization ratio. All data is presented as box plots show the minimum, 1st quartile, median, 3rd quartile, and maximum. All data pooled from three independent biological replicates for each condition. (Nonparametric independent samples Kruskal–Wallis test)

231 and MCF10CA1a cells. This observation is compatible with iEFs preventing formation of actin-rich leading-edges. The present result is also consistent with previous reports for SCP2 cells[11], where cytoplasmic F-actin distribution was diffuse and filopodia formation was suppressed for iEF(+) even with a different cell line. However, iEFs did not alter the actin distribution of the normal MCF10A breast cells. Involvement of focal adhesion kinases (FAK) in actin dynamics for iEF(+) were also explored. Changes in FAK activation were absent for iEF(+) conditions for MDA-MB-231 and MCF10A cells. However, for MCF10CA1a with iEF(+)/EGF(+), partial changes in FAK activation were observed with respect to migration speeds and actin distribution (Supplementary Figs. 7A–I and 8).

**Akt signaling mediates directional responses to iEFs.** Akt signaling is an important pathway regulating the tumor promoting

properties of cells, including motility[32,33]. We hypothesized that the Akt pathway is critical in enabling cells to directionally sense applied iEFs by intracellular electrotransduction. To investigate the role of Akt in iEF sensing, we inhibited Akt phosphorylation using MK2206. MK2206 is a pan-Akt inhibitor[33–35] that potently inhibits phosphorylation of Akt-1, −2, and −3. Application of MK2206 alone on MDA-MB-231 cells had no significant ($p = 1.000$) effect on mean migration speeds compared with controls (Fig. 7a). Moreover, co-application of MK2206 with iEFs (parallel and antiparallel directions) resulted in a comparable modest decrease in average migration speeds compared with untreated controls (Fig. 7a). Therefore, Akt inhibition completely abrogated the ability of MDA-MB-231 to migrate faster with standalone antiparallel iEF (Fig. 7a). MK2206 treatment alone also significantly ($p < 0.001$) reduced the persistence of the MDA-MB-231 cells by 38% (Fig. 7b) compared with untreated controls. However, co-application of MK2206 with antiparallel iEFs (but

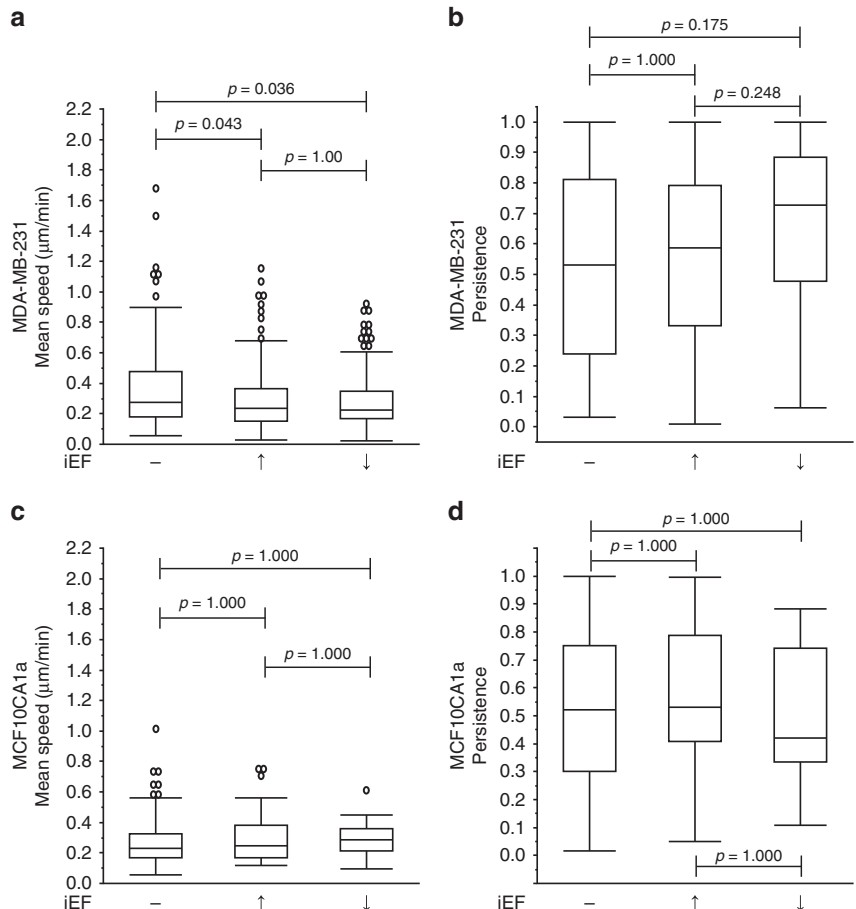

**Fig. 7** Inhibition of Akt phosphorylation impairs the ability of breast cancer cells to sense directionality of iEFs. **a** MDA-MB-231: treatment with MK2206 (2.5 µM) nullified pro-migratory stimulatory effect of antiparallel iEFs. **b** MK2206 treatment also nullified pro-migratory stimulus of antiparallel iEFs on cell persistence. **c** MCF10CA1a: MK2206 treatment had a significant effect on their migration speeds and also resulted in stimulatory effects of antiparallel iEFs on spontaneous migration. **d** MK2206 treatment also nullified the effects of iEFs on cell persistence. The normal MCF10A cells do not migrate under these conditions and iEFs have no effect on the migratory behavior of these cells under these set of conditions. All data is presented as box plots show the minimum, 1st quartile, median, 3rd quartile, and maximum. All data pooled from three independent biological replicates for each condition. (Nonparametric independent samples Kruskal–Wallis test)

not parallel iEFs), had no effect on the persistence of these cells compared with MK2206(+)/iEF(−) (Fig. 7b). In both cases, persistence remained below untreated controls. Akt inhibition thus abrogated directional response (in average migration speeds) of MDA-MB-231 (Fig. 7a). Western blot data showed iEFs had no effect on Akt phosphorylation and total Akt levels (Supplementary Figs. 9 and 10). This implies that Akt phosphorylation is necessary for sensing the direction of applied iEFs, but that iEF treatment confers no direct effects (adverse or stimulatory) on Akt activation and expression.

For MCF10CA1a cells, standalone treatment with MK2206 significantly ($p < 0.001$) decreased median of the mean migration speeds (by ~45%) (Fig. 7c) and persistence, compared with untreated controls. This outcome clearly indicates that Akt signaling mediates even spontaneous (EGF(−)) motility of MCF10CA1a cells. No additional effects on migration speeds were observed with iEF(+) (parallel or antiparallel) and MK2206(+). Of importance, MK2206(+)/EGF(−) treatment successfully abrogated the pro-migratory effect of parallel iEFs on the spontaneous motility of MCF10CA1a (Fig. 2c, d). As with migration speeds, standalone (i.e., iEF(−)/EGF(−) MK2206(+)) treatment significantly ($p = 0.002$) reduced their persistence by ~14% (Fig. 7d) compared with untreated controls. Akt inhibition curbed the

stimulatory effects of iEFs on persistence of MCF10CA1a (for EGF(−) (see Fig. 2d)). iEF(+) did not change Akt phosphorylation or expression levels (Supplementary Figs. 9 and 10). However, the western blots clearly show increased basal level of Akt phosphorylation for MCF10CA1a compared with MDA-MB-231, suggesting that Akt signaling plays a vital role in spontaneous motility for MCF10CA1a cells.

In summary, Akt activation is necessary for breast cancer cells to sense the direction of applied iEFs. Pro-migratory effects of antiparallel iEF(+)/EGF(−) treatment of malignant breast cancer cells were completely abrogated by inhibiting Akt phosphorylation. Of importance, iEFs did not alter the state of Akt signaling. Therefore, either Akt or another molecule downstream in the intracellular Akt signaling cascade plays a vital role in how iEFs affect cell motility.

**Combined effects of iEFs and Akt inhibition on EGF-promoted migration**. To further investigate the hindering effects of iEFs on EGF-promoted migration of breast cancer cells, we blocked Akt signaling, an important downstream effector of EGFR phosphorylation[36,37]. For MDA-MB-231 migrating under EGF(+), MK2206 treatment reduced median of the mean migration speeds

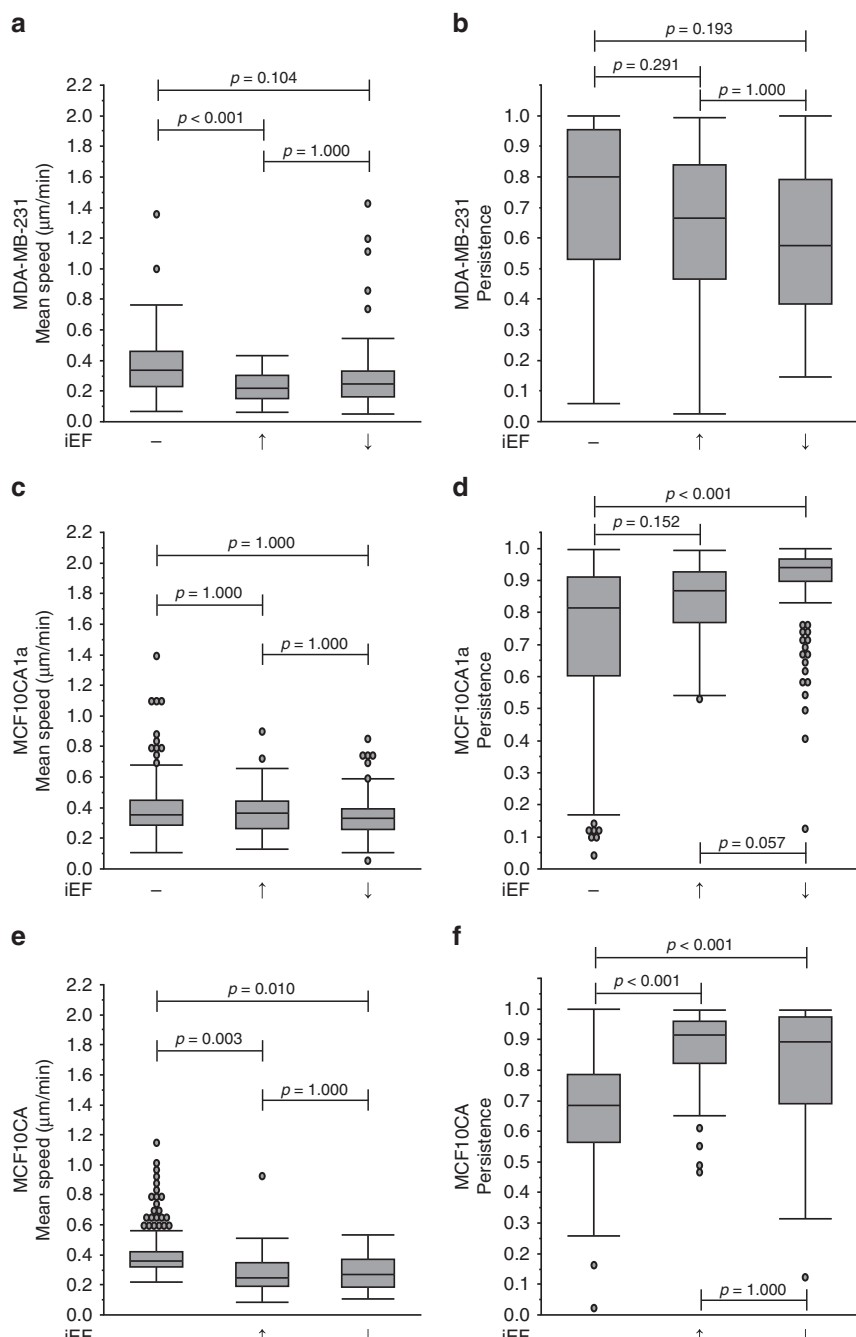

**Fig. 8** Co-treatment with iEFs and MK2206 inhibits EGF-gradient promoted motility. **a** MDA-MB-231: parallel iEFs and MK2206 work together to inhibit EGF-promoted migration speeds below levels of untreated controls. **b** iEFs and MK2206 work together to inhibit cell persistence significantly below levels of untreated control levels. **c** MCF10CA1a: co-treatment with iEFs and MK2206 significantly reduced migration speeds well below levels of untreated controls. **d** Co-treatment of MK2206 with iEFs increased persistence of these cancer cells. **e** MCF10A: co-treatment with iEFs and MK2206 inhibited EGF-promoted motility of normal breast cells. **f** The co-treatment resulted in increase in cell persistence compared with standalone MK2206 treatment. All data are presented as box plots show the minimum, 1st quartile, median, 3rd quartile, and maximum. All data pooled from three independent biological replicates for each condition. (Nonparametric independent samples Kruskal–Wallis test)

by 13% (0.38–0.33 μm min$^{-1}$) returning them to near control levels (Fig. 8a). Co-application of MK2206 with iEFs further reduced median of the mean migration speeds by 42% (parallel iEF) and 34% (antiparallel iEF) below EGF controls, respectively. Both treatments brought cell speeds below untreated control levels (0.22 μm min$^{-1}$ and 0.25 μm min$^{-1}$, respectively) with parallel iEFs resulting in the slowest migration speed observed in this work. As with EGF(−), the EGF(+) results show that

directional cellular response is Akt-dependent. Thus, the combination of iEFs(+)/MK2206(+) more effectively hinders EGF-promoted motility compared with iEF(−)/MK2206(+) or iEF(+)/MK2206(−). Interestingly, Akt phosphorylation and total Akt levels were unaffected with EGF(+) for MDA-MB-231 (Supplementary Figs. 9 and 10). Therefore, iEFs do not modulate migration speeds of EGF-treated MDA-MB-231 cells by directly altering Akt activation or expression.

MK2206(+)/iEF(−)/EGF(+) alone significantly ($p < 0.001$) reduced persistence levels 14% below MK2206(−)/iEF(−)/EGF(+) (Fig. 8b). Combined treatment MK2206(+)/iEF(+) further reduced the persistence of MDA-MB-231 regardless of iEF direction. iEFs reduced persistence by 24% (parallel direction) and by 33% (antiparallel direction) below MK2206(−)/iEF(−)/EGF(−), respectively.

MK2206(+)/iEF(−) treatment of MCF10CA1a cells significantly ($p < 0.001$) reduced EGF-gradient promoted increase in migration speeds, by ~32% (Fig. 8c). Unlike MDA-MB-231, co-application of MK2206(+)/iEF(+) did not additionally inhibit migration speeds, regardless of iEF direction. Migration speeds remained below those of untreated controls. We found no evidence indicating that iEFs change Akt phosphorylation levels or expression in MCF10CA1a cells for EGF(+) (Supplementary Figs. 9 and 10). Hence, it is apparent that Akt signaling is crucial for controlling MCF10CA1a motility.

For normal MCF10A cells (Fig. 8e), MK2206(+)/EGF(+)/iEF(−) treatment significantly ($p < 0.001$) decreased median of the mean migration speeds by ~27% compared with MK2206(−)/EGF(+)/iEF(−). Combined MK2206(+)/EGF(+)/iEF(+) treatment regardless of iEF direction further suppressed mean migration speeds by another ~25%. Therefore, the combined MK2206(+)/iEF(+) treatment again provided a great benefit in reducing cell migration speed. MK2206(+)/EGF(+)/iEF(−) treatment also decreased persistence by ~27% (Fig. 8f) but with MK2206(+)/EGF(+)/iEF(+), persistence returned to those levels corresponding to MK2206(−)/EGF(+)/iEF(+).

## Discussion

Despite extensive investigations over two centuries, it is only recently that the nuances of how migrating cells respond to electrical signals are beginning to be understood[9,11,38]. Previous reports have tacitly conflated the distinct effects of EFs and electric currents, and it is with non-contact application of EFs that this distinction is being elucidated[11]. Here we showed the selective action of iEFs in two triple-negative metastatic cell lines (MDA-MB-231, MCF10CA1a) and contrasted them with effects on nontransformed epithelial breast cells (MCF10A). iEFs provide access to the interior of cells (since magnetic fields can penetrate cell interiors while EFs applied using contact methods cannot penetrate the electrically insulating cell membrane)[39], and our results show differences in electrical characteristics of the cytoplasm depending on cell type and lineage.

A key finding of this work is that antiparallel iEFs increased average speed and persistence of MDA-MB-231 and MCF10CA1a for EGF(−). We also showed that parallel iEFs potently hinder EGF(+) motility of MDA-MB-231, while iEFs, regardless of direction, lower EGF(+) induced increases in MCF10CA1a migration speeds, which in both cases returned to the levels of untreated controls. Thus, iEFs were extremely effective in nullifying the pro-migratory effects of EGF(+) on these triple-negative breast cancer (TNBC) cells. Our results clearly show context dependent pro-migratory and anti-migratory effects of iEFs on TNBC cell migration. Previous reports with dcEFs showed that MDA-MB-231 migrated toward the anode, and that electric currents result in higher speeds compared with untreated controls[40,41]. Of importance, these previous studies showed that reversing direction of electric currents drove cells toward the new anode, clearly showing that the external electric currents associated with dcEF are responsible for directing MDA-MB-231migration and speeds, rather than the EFs driving those electric currents[40]. In contrast, iEFs either have no effect on cell persistence or increase persistence and do not dictate the direction of migration. Moreover, iEFs either have no

effect on directional migration or strengthen the breast cancer cells' ability to maintain migratory directionality. Unlike dcEFs where speeds only increase compared with controls, iEFs have two distinct effects (pro-migratory and anti-migratory) on motility. Therefore, iEFs affect cancer cells differently than the traditional manner of galvanotaxis (where electric currents are driven through the media) and under a new realm of electrotransduction. We define electrotransduction as the conversion of extracellular electrical cues in the absence of electric currents into intracellular biochemical signaling, resulting in distinct cell responses (altered motility, proliferation, apoptosis, metabolism etc.). Our work though is limited to understanding motility changes in breast cancer cell migration.

This new realm of electrotransduction is particularly interesting with regard to MCF10As, which need external cues (chemokine gradients, mechanical microenvironment, etc.) to migrate[42]. iEFs had no effect on these cells, which maintained their non-migratory phenotype irrespective of iEF direction. In contrast, MCF10As exhibit anodal migration with electric current in dcEFs[25]. These contrasting responses to EFs (induced field) and electric currents (direct current, conflated in existing literature with EFs) is strong evidence that an electrotransduction-based sensing mechanism exists for cancer cells with iEFs. This represents an opportunity for devising selective and targeted treatment. EGF(+) induce a migratory phenotype in MCF10As. Antiparallel iEFs stimulated their pro-migratory effect for EGF(+). This mirrored the response of breast cancer cells migrating with iEF(+)/EGF(−). By implication, EGF(+)/iEF(+) transformed MCF10As to closely match the EGF(−) breast cancer cell migratory phenotype. Had these distinct responses to iEFs been based on traditional galvanotactic characteristics, these cell lines would have shown anodal migration. However, we find no evidence supporting these previous reports.

Unlike dcEFs, where electric currents cannot penetrate the electrically insulating cell membrane, iEFs have direct access to the cell interior since they are produced by time-varying magnetic fields, which pass unimpeded through the magnetically transparent cell membrane[39]. Alternating EFs (10 kHz to 1 MHz) have been thought to not have any biological effects[39,43–47] due to their inability to cause net depolarization or result in significant dielectric losses (no heating). This work clearly shows significant effects of alternating EFs on intracellular cell signaling and motility.

To our knowledge, this work is the first to clearly establish a direction-dependent and cell line-dependent response to EFs. Previous reports considered galvanotactic directional migration partly controlled by Akt signaling[48–50], and hence we examined the role of Akt signaling in sensing the net direction of asymmetric iEFs. Inhibition of Akt activation with MK2206 nullified the pro-migratory effects of antiparallel iEFs on EGF(−) spontaneous migration of breast cancer cells, and no changes were observed in Akt activation by iEFs (Supplementary Figs. 9 and 10). Therefore, Akt activation was necessary for sensing the net asymmetry of iEFs for the pro-migratory response of antiparallel iEFs, though the latter did not affect Akt activation or expression. The motility of the cancer cells returns to their base levels when the sensing pathway is removed or the iEF sensing and coupling mechanism is disrupted. The primary control mechanism of iEFs altering cell migration must therefore be by electrotransduction.

The observed suppression of migration speeds and persistence of MDA-MB-231 with MK2206(+)/iEF(+) may be understood as follows. Either iEFs increased the efficacy of MK2206, or iEFs and MK2206 acted concurrently along two independent pathways to suppress the average speed and persistence. Parallel iEFs nullified the effects of EGF gradients on MDA-MB-231 cells and iEF(−)/MK2206(+) treatment also nullified EGF-promoted motility.

However, iEF(+)/MK2206(+) additively suppressed motility to levels below those of controls, suggesting that iEFs and MK2206 are operating in parallel to produce this potent inhibitory response. The motility of MCF10CA1a was markedly suppressed by MK2206 due to its regulation through Akt phosphorylation even in the absence of EGF. MK2206(+)/iEF(+) did not however additionally hinder their motility. These differences between MDA-MB-231 and MCF10CA1a cells confirm that susceptibility to iEFs is cell line dependent.

Non-contact electromagnetic therapy known as TTFs has recently been shown to significantly increase survival outcomes in glioblastoma patients[12,13,51–55]. TTFs operate in a range of frequencies (100–300 kHz)[12,13,51–56] similar to our work (100 kHz). However, TTFs reportedly only affect dividing cancer cells and arrest cell proliferation, with significantly higher EF strength ($\geq 1$ V cm$^{-1}$) while no cell death is observed in our much weaker iEFs ($\sim 100$ µVcm$^{-1}$). To our knowledge, this study is the first of its kind to show that the pro-migratory stimulus of EGF gradients on breast cancer cells can be extrinsically nullified using weak electromagnetic fields. Given the clinical benefit of TTFs both as a standalone and a combinational therapy (with an anti-cancer drug/chemotherapy), our work shows that even weaker electromagnetic fields could specifically target and treat metastatic cancerous lesions from spreading, potentially with no adverse effects.

In summary, we have shown that iEFs alter the motility of metastatic breast cancer cells in directionally-dependent and cell line dependent manner. We also show that iEFs potently decrease migration speeds and migration numbers of metastatic breast cancer cells with EGF(+). iEFs inhibit EGFR phosphorylation in EGF-treated cancer cells thereby altering actin cytoskeleton structure and critical processes involved in cell motility—a potential mechanism explaining the observed hindrance of cancer cell motility by iEFs. Moreover, inhibition of Akt phosphorylation nullifies the directional response of EGF(−)/iEF(+) breast cancer cell motility but confers additional inhibition for EGF(+)/iEF(+)—showing the potential therapeutic value of iEFs when combined with anti-cancer drugs. Importantly, iEFs did not affect migration of normal non-migratory cells for EGF(−), indicating that normal cells in our body remain unaffected with iEF treatment. This work underscores some key attributes of metastasis. First, the extent of metastasis as determined by the number of cells migrated, may or may not be fast. Second, the speed of metastasizing cells, may or may not be accompanied by large numbers. Both attributes are important in clinical significance and may dictate different diagnostic and therapeutic strategies. Our results are significant in identifying how low-frequency (<300 kHz) iEFs interact with mammalian breast cancer cells and in deciphering the governing mechanisms controlling their migratory responses. This work lays the foundation for exploring new non-contact electromagnetic therapies.

## Methods

**Cell lines and reagents**. MDA-MB-231 breast adenocarcinoma cells stably expressing GFP[57] (provided by Luker Lab, University of Michigan, Ann Arbor) were cultured in DMEM (Life-Technologies, 11995073) supplemented with 1% penicillin–streptomycin–glutamine (100 µg mL$^{-1}$, Life Technologies, 10378016), and 10% fetal bovine serum (FBS) (Atlas Biologicals, EF-0500-A, E27D17A1). MCF10A cells (gift from Ostrowski Lab, The Ohio State University, Columbus, OH) were cultured in DMEM-F12 (Corning, 10-103-CV) supplemented with 1% penicillin–streptomycin (100 µg mL$^{-1}$, Life Technologies, 15140122), and 5% horse serum (Life-Technologies, 16050-122, 1783307). The culture media for MCF10A cells was also supplemented with 0.1% insulin (10 µg mL$^{-1}$, Sigma-Aldrich, I1882), 0.05% hydrocortisone (0.5 mg mL$^{-1}$, Sigma-Aldrich, H0888), 0.02% epithelial growth factor (20ng mL$^{-1}$, Peprotech, AF-100-15), and 0.01% cholera toxin (100 ng mL$^{-1}$, Sigma-Aldrich, C8052). The third cell line, MCF10CA1a breast invasive ductal carcinoma cells (Dr Ramesh Ganju Lab, The Ohio State University,

Columbus, OH) were cultured in DMEM-F12 (Life-Technologies, 11330032) supplemented with 1% penicillin–streptomycin (100 µg mL$^{-1}$, Life Technologies, 15140122), and 5% horse serum (Life-Technologies, 16050-122, 1783307). In all our experiments, the culture media was supplemented with 0.1% FBS instead of 10% FBS. We refer to this media as migration media. EGF (25 ng mL$^{-1}$) or Akt inhibitor MK2206 (2.5 µM) were supplemental additions to this media based on the desired experimental condition.

**Helmholtz coil**. Design: A Helmholtz coil was in-house custom designed to accommodate application of iEFs using a Nikon Eclipse TE2000-U microscope (Nikon Instruments Inc.) to generate time-lapse images of cells located inside of a six-well culture plate (Fig. 1a–c). The frame of the coil was designed to fit into the same multiwell plate holder already fabricated for the microscope stage. The condenser of the microscope limited the vertical range of the coil while the focal length of the objective limited the thickness of the coil. In addition, in order to visualize the cells inside the wells, sections of the coil were separated to create windows in order to image the six wells. This required gaps between windings leading to the implementation of a Helmholtz style coil, as can be seen in Fig. 1. The rectangular cross section of the coil was designed so that a multiwell plate could be easily inserted in its bore. With these constraints, the final designed coil comprised four individual segments in a series with a separation of 12 mm for the viewing windows. Each segment measured 22 mm in height and 91 mm in width. The depth of the outer and inner segments were 18 mm and 28 mm, respectively (Supplementary Fig. 11B). Each coil segment had ~10 layers with the outer (Supplementary Fig. 11A, Sub-Coil 1 and Sub-Coil 4) segments having 25 turns per layer and the inner (Supplementary Fig. 11A, Sub-Coil 2 and Sub-Coil 3) segments having 40 turns per layer.

Fabrication: The frame of the coil was designed in SolidWorks and 3D printed with acrylonitrile butadiene styrene as the material. The coil was wound by hand with 32 AWG insulated copper wire (0.202 mm diameter). The turns were separated to try to prevent proximity effects from increasing the AC impedance of the coil. The ends of the wound wire were soldered to a BNC cable with wire leads.

Characterization: The coil inductance and resistance were measured using an LCR meter (Keysight U1733C) at 100 kHz. The capacitance was inferred from measurement of the resonant frequency of the coil as determined from the frequency response[11]. A simple circuit element model of the coil was developed with measured and calculated parameters[11]. This model was used to infer the conduction current through the Helmholtz coil. Using the geometry of the coil, the vector potential was calculated versus position and time, and analytically differentiated with respect to time to calculate the iEF. The magnetic induction (**B**) was calculated using the curl of the vector potential, A and the calculated conduction current in the coil, $I(t)$. The calculated values of the magnetic induction were validated against measurements of **B** using a fluxgate sensor (Magnetic Sciences, Model #MC162). The sensor was placed at the center of the coil and the magnetic induction trace was recorded on an oscilloscope (Agilent DSO-X 2014A). The peak field is ~100 µV cm$^{-1}$, which is at least four orders of magnitude smaller than previous studied. Since power dissipation is given by $\sigma E^2$, where $\sigma$ is the conductivity of the media and $E$ is the electric field strength, the power dissipation with our fields is at least $10^8$ orders of magnitude smaller than other studies, therefore, heating is never an issue with our setup.

**The MBDM assay**. Design: The MBDM assay was designed to have three ports separated by 700 µm long arrays of parallel microtracks (Fig. 1d). The dimension of each port was 50 mm × 15 mm. Cells were seeded in the center port and the top and bottom ports were designated as cell collection port and/or chemokine source reservoir port depending on the experimental condition. Microchannels were designed to have a square cross section of 20 µm × 20 µm. The cross-section dimensions were on the same order as the size of single cells and mimic the dimensions of preexisting microtracks available to cells in-vivo[42]. Moreover, these migratory tracks are representative of physiologically relevant matrix metalloproteinase independent cancer cell migration mode during metastasis[1,2,20,58]. The bi-directional design of this assay allows cells to migrate in either direction from the seeding port and provides a better understanding and quantification of the directional bias of external cues such as chemokine gradients and the directional effects of applied iEFs. The large ports for cell seeding ensure uniform seeding density, excellent cell viability, and repeatability.

Fabrication: The designs for transparency masks were created using AutoCAD-2014 and the final masks were printed at 25000 DPI (CAD/Art Services, OR). A standard photolithography process[59–62] was used to fabricate the silicon masters, wherein a 20 µm thick layer of SU-8 2025 (Spin Speed: 3000 rpm; Spin Time: 90 s) was spin coated on a piranha cleaned test-grade silicon wafer (University Wafer). The coated wafer was then exposed to UV light through the transparency mask, which resulted in crosslinking of the photoresist imprinting the design on the wafer. We treated the exposed wafers with SU-8 developer that washed away the soft uncross-linked SU-8 resulting in formation of the negative pattern of the required micro-channel geometry on the wafer. The wafer was then cleaned with isopropyl alcohol solution and passivated for 30 min in a fume hood with tridecafluoro-1,1,2,2-tertahydrooctyl)−1-trichlorosilane (United Chemicals Ltd, T2492-KG). Salinization passivates the wafer surface and prevents it from sticking to the polydimethylsiloxane (PDMS). All the processing until this stage was done in

a Class 100 Cleanroom. A technique called replica molding was used to get the final microtrack based migration assay from the silicon master[60]. A 10:1 solution of PDMS Base Elastomer and Cross-linker (Sylgard 184 Silicone Elastomer, Dow Corning Corporation) was poured over the wafer, degassed, and cured at 65 °C for 2 h. Cured PDMS was peeled off the silicon master, and was cut into 20 mm × 20 mm square pieces. For fabricating the seeding and the collection ports in the devices, we punched holes using a 4 mm biopsy punch; these devices were then plasma oxidized and irreversibly bonded to cured PDMS in six-well culture plates. The six-well culture plate was sterilized in high-intensity UV light and each device was treated with $10 \mu g \, mL^{-1}$ of fibronectin and incubated at 37 °C for 90 min; PDMS absorbed the fibronectin and made the surface conducive for cell attachment and growth.

EGF-gradient characterization: To characterize the biomolecular gradient profile in the MBDM assay, 10 kDa FITC conjugated dextran was used as a surrogate fluorescent tracer for EGF which has a molecular weight of ~6 kDa[63]. FITC-dextran was prepared in 1× phosphate buffer saline (PBS) to a concentration of $1 \, mg \, mL^{-1}$. The seeding and the bottom collection ports on the device were filled with 1× PBS and the top port (in this case the chemokine port) was filled with $1 \, mg \, mL^{-1}$ FITC-Dextran solution while ensuring that no fluid flow takes place from top port to the middle or bottom port in order to establish purely mass transfer (diffusion) based gradients. The device was then monitored under a stereo microscope (Nikon Instruments Inc.) for 12 h at intervals of 5 min between each frame (Supplementary Fig. 2A). Gradient profiles and diffusion coefficients were then quantified using both finite volume numerical simulation in COMSOL Multiphysics 5.3a and NIH ImageJ Image Processing Software (Supplementary Fig. 2B–H, Supplementary Equations 7–9).

Cell seeding and migration: Cells were washed in culture plates with 1× PBS solution, treated with 0.05% trypsin-EDTA solution (Sigma-Aldrich), and then counted using a hemocytometer. Cells ($2 \times 10^5$) suspended in migration media were seeded in the middle port (seeding port) with an extremely small flow from the collection ports to the seeding port, which equilibrated in <15 min. The cells inside the devices were then incubated for 12 h in migration media following which the media was aspirated and replaced with new media based on the experimental conditions. For experiments with growth factor stimulation, the migration media was supplemented with EGF ($25 \, ng \, mL^{-1}$) and was introduced in only one of the two collection ports. Devices were incubated for another 36 h in culture media and refreshed every 12 h. Cell migration was then observed using a time-lapse scope in a live-cell chamber for 12 h. In the case of experiments involving the Akt inhibitor, the media was supplemented with $2.5 \mu M$ of MK2206 immediately before the 12-h time-lapse.

**The transwell migration assay**. Transwell permeable supports that have 6.5 mm diameter inserts of polycarbonate membrane with 8-μm-diameter pore size (Corning, CLS3422) were used in the experiments reported here. Each Transwell insert was coated with $80 \mu L$ of $10 \mu g \, mL^{-1}$ fibronectin solution (in 1× PBS, Corning Inc., 354008) and left to dry for 12 h. Cells were simultaneously serum starved in migration media for 12 h. Following this step, the cells were removed using 0.05% trypsin-0.02% EDTA solution (Sigma-Aldrich, 59417 C) and handled exclusively in migration media. Cell suspensions with a concentration of $1 \times 10^6$ cells $mL^{-1}$ were prepared and $150 \mu L$ of this media ($1.5 \times 10^5$ cells) was plated in the top chamber of each Transwell insert and the bottom chamber was filled with $600 \mu L$ of migration media or EGF supplemented migration media ($100 \, ng \, mL^{-1}$ EGF). After 8 h, the cells were fixed using a standard HEMA 3 solution kit (Fisher Scientific, 23-123869) and imaged using a Stereo Microscope (Leica Microsystems Inc.). Cell migration numbers were then quantified using a custom MATLAB script as described in Image Acquisition and Processing. Cell migration without growth factors and without iEFs served as controls and all the other conditions were normalized with respect to these controls.

**Immunofluorescence**. Imaging of F-actin in the microfluidic bi-directional microtracks assay: Cells in the devices were fixed in 3.7 (wt vol$^{-1}$) paraformaldehyde solution for 30 min. They were subsequently washed with 1× PBS three times. For F-actin labeling, we blocked the cells for 60 min in blocking buffer (0.1% Triton-X and 5% donkey serum in 1× PBS). Cells were then treated with Alexa Fluoro® 488-conjugated or 555-conjugated phalloidin for 60 min (1:40, Thermo-Fisher), again followed by a 1× PBS wash (three times, 10 min each). Finally, the nuclei were labeled with 4′,6-diamidino-2-phenylindole (DAPI) (1:5000, Sigma-Aldridge, D9542). The nuclei labeling was followed by a final 1× PBS wash (three times, 10 min each) and the devices were left overnight in 1× PBS at 4 °C.

Immunofluorescence staining of EGFR and F-actin: In these experiments, $1 \times 10^4$ cells were cultured on fibronectin-coated ($10 \mu g \, mL^{-1}$), 22 mm #1 glass slides. Cells were allowed to adhere for 12 h in growth media. Cells were then serum starved in migration media for another 12 h. The media was then replaced with fresh migration media. For EGF treated cases, the migration media was supplemented with $25 \, ng \, mL^{-1}$ of EGF. Cells were then either incubated with or without iEFs for 12 h. The cells were fixed with 3.7% (wt vol$^{-1}$) paraformaldehyde solution for 20 min and then washed three times with 1× PBS (5 min each). We then blocked the cells for 60 min using a blocking buffer (0.1% Triton-X 100, 5% goat serum in 1× PBS). The cell samples were treated with the primary EGFR antibody (1:1000, MA5-13319, Thermo-Fisher, diluted in blocking buffer) and left

overnight at 4 °C. We then washed the cell samples three times with 1× PBS supplemented with 0.1% Tween-20 (1× PBST) for 15 min each. We then added the secondary antibody (Anti-Rabbit Alexa Fluro®488, 1:2000 in blocking buffer) and stored the cells in the dark at room temperature for 60 min followed by three 1× PBST washes for 15 min each. We used ActinRedTM 555 ReadyProbes® Reagent (Thermo-Fisher) based on manufacturer instructions to tag the F-actin cytoskeleton. We then washed the samples three times with 1× PBST for 15 min each. Finally, the cell nucleus was stained with DAPI (Sigma-Aldrich, 1:5000 in DI water, D9542) and samples were washed three times with 1× PBST for 15 min each. We then mounted the samples using Fluoromount-G® (Southern Biotech), let them dry overnight at room temperature, and then imaged them using the LSM 700, a high-resolution laser scanning confocal microscope (ZEISS Instruments Inc.).

**Western blot**. For these experiments, we plated $1 \times 10^6$ cells per well in six-well plates in growth media for 12 h followed by migration media for another 12 h. Fresh migration media was added to the top three wells in each plate (1, 2, and 3), while EGF ($25 \, ng \, mL^{-1}$) and supplemental migration media were added to the bottom three wells (4, 5, and 6). One of the cell-containing six-well plates was then treated with iEFs for 12 h. Immediately after treatment, the plates were placed on ice and each well was then washed three times with 1× tris-buffered saline (TBS, Corning Inc.) solution. TBS was aspirated out and 1 mL ice-cold radio-immunoprecipitation assay (RIPA) buffer, supplemented with a protease inhibitor and a phosphatase inhibitor, was added to each well. Cells were scraped out using a cold plastic cell scraper and the cell suspension was transferred into a precooled microcentrifuge tube. The cell suspension was then spun at 15,000 rpm for 20 min in a 4 °C precooled centrifuge. The centrifuge tubes were placed on ice and the supernatants were transferred to fresh tubes and kept on ice. The pellet at the bottom of each microcentrifuge tube was discarded. Protein in each tube was estimated against a standard bovine serum albumin (BSA) solution ($1.42 \, mg \, mL^{-1}$) using the DC$^{TM}$ Protein Assay Kit II (Bio-Rad, 500-0112). We then collected 50 μg of total protein mixed with 10 μL dye (Invitrogen, NP0007) and 5 μL reducing agent (Invitrogen, NP0009) from each condition and loaded it on to a 4–12 gradient gel (Invitrogen, NP0335BOX). We then placed the gel in the running buffer (Invitrogen, NP0001, 1:20 dilution in DI water) at 120 V for ~2 h. Gels were then placed in a transfer buffer (Tris/Glycine Buffer, Bio-Rad, 161-0771––diluted to 1x with 20% methanol in DI water) for 5–10 min following which the transfer sandwich was prepared. The sandwich was placed in a transfer tank and run at 18 V for 90 min. The blot was then washed with 1× TBST (1× TBS with 0.1% Tween-20) three times for 15 min each. The blot was blocked with 5% BSA in TBST solution for 1 h at room temperature. Primary antibodies p-EGFR (1:1000, Cell Signaling Technology, 4407S, 3777S), p-Akt (1:2000, Cell Signaling Technology, 9271S), p-FAK (1:1000, Thermo-Fisher, 700255), t-EGFR (1:200, Santa Cruz, sc-03-G), t-Akt (1:200, Santa Cruz, sc-8312), t-FAK (1:1000, Cell Signaling Technology, 3285 S), and GAPDH (1:1000, Cell Signaling Technology, 5174S) were then prepared in the blocking solution (5% BSA in TBST) and left overnight on a rocker at 4 °C. We washed the blots three times for 15 min each with 1× TBST solution. Secondary antibody (1:2000, GE Healthcare, LNA934V/AH) was prepared in the blocking solution and the blots were treated with the secondary antibody solution for 2 h at room temperature followed by three 15-min washes with TBST solution. Blots were then treated in the dark with the Pierce® ECL Western Blotting Substrate (Thermo Scientific, 32209) for 5 min and then developed using standard solution in an X-ray room based on protocol provided by the manufacturer.

**Image acquisition and processing**. Time-lapse movies: The time-lapse movies were acquired using a Nikon Eclipse TE2000-U microscope (Nikon Instruments Inc.) in 5-min intervals between images for 12 h using a 10x objective (Fig. 1b). The on-stage incubator maintained $CO_2$ levels at 5% and the temperature at 37 °C for the duration of the experiment. The time-lapse movies were analyzed using the MtrackJ plugin[64] in Fiji[65] to determine average cell speed, distance traveled, and displacement data.

Transwell migration assay: Images from experiments using the Transwell migration assays were taken using a stereo microscope (Leica Microsystems Inc.) after fixation and staining of the cells using the HEMA 3 solution kit (Fisher Scientific). Images were taken at a magnification such that the entirety of the Transwell membrane was within the frame. The images were then imported into MATLAB and analyzed using a custom code (MATLAB script) (see Supplementary Information). The script splits the images into RGB components and uses the Otsu method[66] for setting the background threshold intensity so that the cell can be distinguished from the background using the inverse of the green channel. All groups of pixels with connectivity of at least eight pixels were identified as single objects. To account for clustering of cells, each object's area was divided by the average area of a cell. The average area of a cell was determined using the mean of manual measurements of ~20 isolated cells for each case. The count from each cluster was rounded to the nearest integer value and summed to obtain the total cell count. The plotted values are all normalized to the control conditions for each case.

Actin immunofluorescence in the MBDM assay: Actin immunofluorescence images were acquired using a Nikon TE200 epifluorescence microscope (Nikon

Instruments Inc.) under a 20× objective. The immunofluorescence images were quantified using custom MATLAB scripts (Supplementary Info). The custom MATLAB script calculated the geometric center for an individual cell, i.e. the arithmetic mean of the locations of all pixels comprising the cell area. Then the distance from this geometric center and angle ($0° \leq \theta \leq 360°$) of every pixel relative to a horizontal axis ($\theta = 0°$) was calculated. The cell was then divided into 360 equal sectors each with a sector angle of 1°. Each 1° sector was defined as an individual bin. Irrespective of the cell shape, it is considered a unit circle for the purposes of this calculation. A moment of intensity is calculated for individual pixels and this value is summed for all the pixels in every individual sector. This total value is normalized to the total number of pixels in that sector. This method is summarized in the following equation:

$$J = \frac{1}{N}\sum_{k=1}^{N} r_k^{\alpha} I_k,$$

where $J$ is the moment of intensity for an individual bin, $N$ is the total number of pixels in the bin, $r_k$ is the distance of pixel $k$ from the centroid, $\alpha$ is a weighting factor (cell aspect ratio), and $I_k$ is the intensity of pixel $k$. Finally, all the 360 individual bins are normalized with respect to the maximum value of the summed moments and the normalized value for each sector is plotted on the unit circle giving a visual and quantitative representation of the distribution of intracellular actin. An index referred to here as the PR, is used to determine whether the intracellular actin is distributed in a preferential way. The PR is defined as the ratio of the number of occurrences of high ($\geq 0.8$) normalized summed moments of intensities in the sectors $75° \leq \theta \leq 105°$ and $255° \leq \theta \leq 285°$, to the total occurrences of high ($\geq 0.8$) normalized summed moments of intensities.

$$PR = \frac{\sum\limits_{\theta=75°}^{105°} N_\theta(\underline{J}(\theta)>0.8) + \sum\limits_{\theta=255°}^{285°} N_\theta(\underline{J}(\theta)>0.8)}{\sum\limits_{\theta=0°}^{360°} N_\theta(\underline{J}(\theta)>0.8)}$$

Where $\underline{J}(\theta)$ is the normalized moment of intensity at the angle $\theta$, $N_\theta(\underline{J}(\theta))$ is the number of bins, and $\sum\limits_{\theta=0°}^{360°} N_\theta(\underline{J}(\theta)>0.8) \geq 1$. A PR of 1 thus implies that all the intracellular F-actin is primarily localized at the leading and/or trailing edges of the cell, whereas a value of 0 indicates no localization at the leading and trailing edges. A PR of 0.167 implies an even distribution of actin throughout the cell. Only single-isolated cells that were migrating inside the channels were analyzed using this approach.

Immunofluorescence staining of EGFR and F-actin: Images were acquired using the LSM700 laser scanning confocal microscope (Carl Zeiss Microscopy GmbH., Germany) with a 63× oil objective. Laser power was controlled by setting the input voltage to the laser source to 5.0 V and the gain was set to 550 (DAPI channel, nucleus), 600 (Alexa Fluor® 488, EGFR), and 600 (Alexa Fluor® 555, Actin) for all samples.

Western blots: Western blot analyses of lysates were performed as previously described[67]. The scanned images of the blots were imported into Fiji[65] and converted into 8-bit grayscale image. A bounding rectangle that enclosed the largest size blot was drawn, and raw intensity was measured for blot under individual condition. The raw integrated intensity was measured and normalized within each condition for each individual protein. This process was also done for the loading control which was GAPDH in this case. Then the ratio of the normalized values for the protein to the normalized value of GAPDH for that condition was calculated.

**Statistics and reproducibility**. We first checked for each case if the distribution of mean cell migration speeds and persistence for different conditions for the three cell lines followed a normal distribution using the Shapiro–Wilk test of normality, where a $p$-value of <0.05 was considered as threshold for considering the dataset to be skewed and not normally distribute. All the data sets for speeds and persistence were found to not follow a normal distribution. This analysis was performed on IBM SPSS Statistics 25 package.

We then compared the sample populations for mean migration speeds and cell persistence by the Kruskal–Wallis test (thereby absolving the requirement for the dataset to be normally distributed) followed by pairwise comparisons in case the null hypothesis was rejected when the $p$-value was <0.05. The test statistic was adjusted for ties. For the MDA-MB-231 and MCF10CA1a cell lines, we had 11 degrees of freedom (dof). In case of MCF10A cell line, we had 5 dof (IBM SPSS Statistics 25).

For the MDA-MB-231 and MCF10CA1a cells, this test resulted in rejection on the null hypothesis that the distribution of the mean migration speeds was same across all conditions ($p = 0.000$). This was followed by post-hoc testing where the significance values had been adjusted by the Bonferroni Correction for multiple tests, where the test statistic was also adjusted for ties. Data points on the box plots represent the minimum value, 1st quartile, median, 3rd quartile, and maximum value for each condition. This was also true for cell persistence for both these cell lines ($p = 0.000$ for Kruskal–Wallis test). The distribution of speeds and cell persistence was found to be different across conditions for MCF10A cells as $p$-value

was reported to be 0.000 for the independent samples Kruskal–Wallis test. This was followed by post-hoc testing where the significance values had been adjusted by the Bonferroni Correction for multiple tests, where the test statistic was also adjusted for ties. Again, a $p$-value of <0.05 was considered as the threshold for statistical significance (IBM SPSS Statistics 25).

Western blot analysis involved one-way ANOVA followed by post-hoc unpaired, two-tailed Student $t$ test. A $p$-value <0.05 was used as the threshold for statistical significance. The data points on the figures represent the mean values and error bars depict standard error in mean (SEM). In all cases, dof for ANOVA was three with $F$-value < 0.0001.

Comparisons for the intracellular actin distribution were done with the independent samples Kruskal–Wallis Test. The null hypothesis that the distribution of the PR was same across all treatments was rejected if the $p$-value was <0.05, which was used as the threshold for statistical significance. This analysis was followed by post-hoc testing where the significance values had been adjusted by the Bonferroni Correction for multiple tests, where the test statistic was also adjusted for ties. Data points on the box plots represent the minimum value, 1st quartile, median, 3rd quartile, and maximum value for each condition. For the MDA-MB-231 cells, a $p$-value of 0.016 was calculated for the independent samples Kruskal–Wallis test with 5 dof. For MCF10CA1a cells a $p$-value of 0.008 was calculated for the independent samples Kruskal–Wallis test with 5 dof. For MCF10A cells a $p$-value of 0.636 was calculated for the independent samples Kruskal–Wallis test with 2 dof (cells do not migrate or enter channels in the absence of EGF gradients). This analysis was carried out using the IBM SPSS Statistics 25 package.

For the transwell assay, the population distribution for different treatments was first tested with the one-way ANOVA test followed by post-hoc Tukey–Kramer HSD method. For the MDA-MB-231 and MCF10CA1a cells, the dof was 5 and the $F$-value was <0.0001. For the MCF10A cells, the dof was 3 and the $F$-value was again <0.0001. For the post-hoc Tukey–Kramer HSD method for pairwise comparison, a $p$-value of <0.05 was used as the threshold for statistical significance. This analysis was done in JMP 14 Pro (SAS).

**Reporting Summary**. Further information on research design is available in the Nature Research Reporting Summary linked to this Article.

## Data availability
All data in support of the findings of this study are available from the corresponding author by reasonable request. Source data underlying graphs are presented in Supplementary Data 1.

## Code availability
Custom codes used for Transwell cell counting, actin polymerization are shown in the Supplementary Information.

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

## Acknowledgements

The authors acknowledge funding from NSF (CBET-1752106), The American Cancer Society (IRG-67-003-50), Pelotonia Junior Investigator Award, NIH (R01HL141941), and The Ohio State University Materials Research Seed Grant Program, funded by the Center for Emergent Materials, an NSF-MRSEC, grant DMR-1420451, the Center for Exploration of Novel Complex Materials, and the Institute for Materials Research. Portions of this work were supported by the Office of the Assistant Secretary of Defense for Health Affairs, through the Lung Cancer Research Program, under Award No. W81XWH-17-1-0233. Opinions, interpretations, conclusions and recommendations are those of the author and are not necessarily endorsed by the Department of Defense. One of the authors (A.A.G.) gratefully acknowledges support as a FAST (Future Academic Scholars Training) Scholar from the Department of Mechanical and Aerospace Engineering at The Ohio State University.

## Author contributions

This study was designed by A.A.G., T.H.J., S.M.M., S.M., D.A., D.S., R.K.G., V.V.S. and J.W.S. Experiments were performed by A.A.G., T.H.J., S.M.B., S.M., J.F., P.K. and D.A.

Data was analyzed by A.A.G., T.H.J., J.F. and P.K. Paper was prepared by A.A.G., T.H.J., S.M., K.K., R.K.G., V.V.S. and J.W.S.

## Additional information

**Competing interests:** The authors declare no competing interests.

