## [Peer Review File · Communications Biology]

Reviewers' comments:

Reviewer #1 (Remarks to the Author):

The manuscript by Garg and colleagues is a well-written characterization of the effects of non-contact induced electric fields (iEF) on the migratory behavior of breast cancer cells in confined channels. Key observations by the authors include a differential response to iEFs among MCF10A, MDA-MB-231, and MCF10CA1a cells. Further, migration of these cells in response to iEF was modulated by the presence of EGF gradients, and iEF seemed to induce spatially biased clustering of the EGF receptor, and the migratory behavior in response to iEF is Akt-dependent. While electro-/galvanotaxis remains a bit controversial in terms of the mechanisms governing the cell response, the present study clearly reports data and potentially provides some insight into signaling mechanisms involved in the response to electric fields. Prior to publication some points should be addressed as outlined below:

1) A graphical representation of the iEF waveform should be provided. Why was only one waveform chosen? Is it possible that there are different responses depending on the magnitude of the net electric field? It seems with this platform, the authors have the capacity to screen the effects of the iEF waveform, which would be informative to other scientists in the field.

2) The polarization of the EGF receptor is an interesting observation. The authors should split the EGFR channel out in grey in Fig. 4A to more clearly see the distribution. Similarly the cell morphologies in Fig. 4 and Fig. 6 are quite different, particularly for the MCF10A cells.

3) The sensing aspect is particularly interesting, given the relationship between chemotaxis and galvano/electrotaxis described here. The integrins are known modulators of chemotaxis, and the role of the matrix is particularly interesting in the context of cancer. Have the authors looked at how modulating ECM coating of the PDMS micro channels affects the response to iEFs?

4) Defining a new term, as the authors do in the discussion: "We define electrotransduction..." implies a response or observation that does not fit contemporary nomenclature. However, it is unclear how 'electrotransduction' differs from the electro-/galvanotaxis response that has been reported for centuries, by the authors own admission. Furthermore, how does this electrotransduction differ from other known electrically-mediated cell signaling such as streaming potential in cartilage as detailed by Grodzinsky and others?

Reviewer #2 (Remarks to the Author):

In their manuscript „Electromagnetic fields alter the motility of metastatic breast cancer cells“, Garg et al. developed a novel microfluidic platform to apply chemical gradients of compounds and weak (<100 $\mu\text{V}/\text{cm}$) asymmetric, non-contact induced electric fields (iEFs). In addition, they evaluated the effect of iEFs in conjunction with EGF gradients and Akt inhibition on the migration of metastatic breast cancer cell lines.

The authors show that there are cell line specific differences in the cells' response to iEFs, that Akt plays a pivotal role for the cells to sense the direction of the applied iEFs and in modulating the migration response. iEFs downregulated EGFR activation and prevented formation of actin rich filopodia in breast cancer cells in the presence of EGF.

The report is very well written, describes an innovative new technique and the results are sound and of importance for the field. Since I am not a physicist, I am not qualified to comment on the construction and characterization of the Helmholtz coil coupled to the microfluidic bi-directional microtrack assay. I am also not qualified to comment on the provided automatic computation codes. Concentrating on the general style of the manuscript and the cell culture data, I have only some minor

criticism.

1. The authors should comment on the differences between the MDA-MB-231 and MCF10CA1a breast cancer cell lines when they first introduce them in line 80. Why did they choose these two cell lines for their experiments?
2. The supplementary figures should be presented in the order of their first mention in the results section, e.g. Fig. S3 is mentioned first in line 108, followed by Fig S2 in line 110 and Fig. S1 in line 121. Therefore, I recommend renaming Fig S3 to S1 and Fig S1 to S3 and showing them in the correct order in the supplementary material.
3. In most of their figures the authors compare the three cell lines MDA-MB-231, MCF10CA1a and MCF10CA, except in Figures 2 and 7. I recommend also showing the "normal" cell line MCF10CA in these figures for consistency, even though they did not migrate as explained in lines 148-155 of the results section - or at least to comment briefly in the figure legend on why they are not shown to avoid confusion of the reader when looking at the figure without having read those lines already.
4. The x-axis is crossing the y-axis at different positions for the different cell lines but at similar positions in various experiments. I assume that this refers to the migration speed and persistence of the respective untreated control cells from Figure 2 as a reference? However, this needs to be explained in the figure legends for a better understanding.
5. In the discussion lines 437, 439 and 454 two tables are mentioned, but there are no tables provided within the manuscript or supplementary files.
6. In lines 524 and 526 the authors refer to TFields as "electromagnetic therapy". TFields are electric fields of low intensity and intermediate frequency. They are not electromagnetic fields.
7. The authors provide a supplementary video, but they do not refer to this video in their manuscript.
8. In the materials and methods section there seems to be some mix-up in cross-referencing to the supplementary figures. In lines 612-614 the authors speak about the measurements of the Helmholtz coil and refer to figure S6. However, Figure S6 shows the effect of iEFs on levels of p-Akt. Probably the authors mean Figure S7? A similar error seems to be in lines 684, 687 and 792/793 when referring to (the wrong) Figures S5 and S6. This requires correction.
9. This reviewer is working on Tumor Treating Fields (TFields). TFields application is inducing heat in the culture dish which needs to be tightly controlled. Do iEFs induce any temperature change within the microfluidic bi-directional microtrack assay and how is this controlled?
10. In the legend of figure 4, line 938 it should read "iEFs have no effect on EGFR distribution..."
11. In the legend of Figure S6, line 2 the word "levels" appears twice.
12. The authors should decide whether they like to put a space character between numerical value and unit or not. This is jumbled throughout the manuscript.

Response to Reviewers' Comments and Questions
Manuscript COMMSBIO-19-0511-T

Response to Reviewer #1:

The authors thank this reviewer for the thoughtful review of our manuscript. In response to the reviewer's questions and comments, we have made revisions to our manuscript. In addition, the reviewer's specific questions and comments are addressed in the following itemized form, and related to revisions in the revised manuscript.

Reviewer's opening comments:

The manuscript by Garg and colleagues is a well-written characterization of the effects of non-contact induced electric fields (iEF) on the migratory behavior of breast cancer cells in confined channels. Key observations by the authors include a differential response to iEFs among MCF10A, MDA-MB-231, and MCF10CA1a cells. Further, migration of these cells in response to iEF was modulated by the presence of EGF gradients, and iEF seemed to induce spatially biased clustering of the EGF receptor, and the migratory behavior in response to iEF is Akt-dependent. While electro-/galvanotaxis remains a bit controversial in terms of the mechanisms governing the cell response, the present study clearly reports data and potentially provides some insight into signaling mechanisms involved in the response to electric fields.

Authors' response:

The authors thank this reviewer for the meticulous and thorough review of our manuscript. In response to the reviewer's questions and comments, we have made revisions to our manuscript. In addition, the reviewer's specific questions and comments are addressed in the following itemized form, and are related to revisions in the revised manuscript.

Reviewer's comment #1:

A graphical representation of the iEF waveform should be provided. Why was only one waveform chosen? Is it possible that there are different responses depending on the magnitude of the net electric field? It seems with this platform, the authors have the capacity to screen the effects of the iEF waveform, which would be informative to other scientists in the field.

Authors' response:

The reviewer is correct that a plot of the imposed voltage waveform should be provided. The waveform used throughout this work was the same sawtooth waveform described in detail in our earlier work (ref. 17 of this manuscript). We have added sentences in the main manuscript describing the sawtooth waveform (also referenced in Fig. S11B in the supplementary information section). In addition, we have added a trace of the 100 kHz, 20 Vpp, sawtooth waveform in Fig. S1D as well. In this manuscript, only one voltage waveform was chosen to drive the current through the coil, so as to be able to benchmark the results reported here against our earlier work (ref. 17). This particular waveform was chosen (as described in ref.17) (1) because of the capabilities of the available equipment (i.e. the function generator, Hewlett-Packard 33120A), and in particular, (2) because of the sawtooth waveform's steep ~50 ns drop off. This sharp drop in the imposed voltage results in a rapidly changing coil current, producing the largest $\partial B/\partial t$ we could generate with our function generator at the maximum input peak-to-

peak voltage of 20V. From Ampere's law, the faster the current in the coil changes, the higher the generated $\partial B/\partial t$ (B being the magnetic induction), and from Faraday's law, the higher $\partial B/\partial t$, the larger the magnitude of the iEF (see Fig. S1, hence the reason for the choice of the sawtooth waveform). The reviewer is correct in pointing out that different responses could result from different magnitudes of the net electric field, and that with this platform, we do have the capacity to explore any possible effects of the iEF waveform. We will explore this opportunity in subsequent work. In summary, we chose the particular waveform with the highest possible $\partial B/\partial t$ and maximum voltage at the frequency corresponding to our earlier work (ref.17) in order to enable comparison with our previous results.

The graphical representation of the iEF waveform and the asymmetry of the iEF has been shown in Supplementary Figure 1 (Fig S1).

Reviewer's comment #2:

The polarization of the EGF receptor is an interesting observation. The authors should split the EGFR channel out in grey in Fig. 4A to more clearly see the distribution. Similarly the cell morphologies in Fig. 4 and Fig. 6 are quite different, particularly for the MCF10A cells.

Authors' response:

We thank the reviewer for suggesting that the EGFR channel be displayed in grey in Fig. 4A in order to make the results more visible. **We have done exactly that in the revised manuscript**, we have added the EGFR channel images corresponding to the images in Fig 4A in the supplementary information (Fig S5).

The cell morphologies for MCF10A cells in Figs. 4 and 6 appear different because the results shown from Fig. 4 are from a culture plate whereas Fig. 6 shows cells migrating in a microchannel.

Reviewer's comment:

The sensing aspect is particularly interesting, given the relationship between chemotaxis and galvano/electrotaxis described here. The integrins are known modulators of chemotaxis, and the role of the matrix is particularly interesting in the context of cancer. Have the authors looked at how modulating ECM coating of the PDMS micro channels affects the response to iEFs?

Authors' response:

The reviewer raises an excellent point with regards to modulating the ECM coating of the PDMS microchannels. We have not explored this line of investigation but hope to do so in future work. We thank the reviewer for this suggestion.

Reviewer's comment:

Defining a new term, as the authors do in the discussion: "We define electrotransduction..." implies a response or observation that does not fit contemporary nomenclature. However, it is unclear how 'electrotransduction' differs from the electro-/galvanotaxis response that has been reported for centuries, by the authors own admission. Furthermore, how does this

electrotransduction differ from other known electrically-mediated cell signaling such as streaming potential in cartilage as detailed by Grodzinsky and others?

Authors' response:

Our understanding of contemporary nomenclature vis-à-vis electrotransduction, is that the term refers to conversion of a sensory input into an electrical signal and vice versa. In the present work, we refer to electrotransduction as the conversion of external cues provided by the iEF into intracellular biochemical signaling leading to several phenotypic responses. Past work on electrotaxis and galvanotaxis (as we point out in ref. 17) is electrochemical in nature due to the presence of externally driven current flow. In the present work, induced electric fields lead to biochemical signaling in the absence of any externally driven current flow. In this sense, our work differs from electromechanical signaling as discussed by Grodzinsky and others. Our work shows clearly that the EGF-EGFR binding event (among others) leads to downstream intracellular signaling that is electrical in nature as it is profoundly affected by iEFs.

The differences between electrotransduction and electro-/galvanotaxis: Paragraph 2 and 3 (lines 434 through 483) of the discussion present the lines of evidence that differentiates traditional electrotactic responses from what we have defined as electrotransduction.

Differences between streaming potential in cartilage vs electrotransduction: In case of cartilage, mechanical deformation and the resulting fluid flow can generate electric fields, and those electric fields can in turn influence the dynamic flow and stiffness properties of the tissue. It is the flow of ions in and out of the cartilage, either due to mechanical deformation or due to application of external currents that result in streaming potentials. Since these potentials are a direct result of induced fluid flow, or a stream of moving fluid, they are called as streaming potentials. This phenomenon is completely physical, requires fluid flow into and out of the cartilage tissue, and can be altered by changing pH of the medium. However, in case of iEFs, the external fields are being sensed (the net asymmetry) and we present direct evidence of these iEFs downregulating EGFR phosphorylation and in turn EGF-promoted breast cancer motility. Therefore, unlike electrokinetic transduction in cartilage, the cancer cells are able to sense and respond to these iEFs that are devoid of any current flow.

Since streaming potentials are a tissue-level phenomena, whereas our results elucidate the ability of individual cells to sense and respond to iEFs, we elected to not elaborate on this point in the revised manuscript text.

Ref: Frank, Eliot H., and Alan J. Grodzinsky. "Cartilage electromechanics—I. Electrokinetic transduction and the effects of electrolyte pH and ionic strength." *Journal of biomechanics* 20.6 (1987): 615-627.

In summary, we have made revisions to our original manuscript taking into account all of the reviewer's helpful comments and criticisms, and further addressed the reviewer's concerns and questions in this response. We believe our revised manuscript is now worthy of publication in *Communications Biology*.

Response to Reviewers' Comments and Questions

Manuscript SREP-14-12761-T

Response to Reviewer #2:

Reviewer's opening comment:

In their manuscript "Electromagnetic fields alter the motility of metastatic breast cancer cells", Garg et al. developed a novel microfluidic platform to apply chemical gradients of compounds and weak ($<100 \mu\text{V}/\text{cm}$) asymmetric, non-contact induced electric fields (iEFs). In addition, they evaluated the effect of iEFs in conjunction with EGF gradients and Akt inhibition on the migration of metastatic breast cancer cell lines. The authors show that there are cell line specific differences in the cells' response to iEFs, that Akt plays a pivotal role for the cells to sense the direction of the applied iEFs and in modulating the migration response. iEFs downregulated EGFR activation and prevented formation of actin rich filopodia in breast cancer cells in the presence of EGF. The report is very well written, describes an innovative new technique and the results are sound and of importance for the field. Since I am not a physicist, I am not qualified to comment on the construction and characterization of the Helmholtz coil coupled to the microfluidic bi-directional microtrack assay. I am also not qualified to comment on the provided automatic computation codes. Concentrating on the general style of the manuscript and the cell culture data, I have only some minor criticism.

Authors' response:

We sincerely appreciate this reviewer's meticulous review of our manuscript, and the positive feedback provided. In response to the reviewer's questions and comments, we have made revisions to our manuscript. In addition, the reviewer's specific questions and comments are addressed in the following itemized form, and related to revisions in the revised manuscript.

Reviewer's comment #1:

The authors should comment on the differences between the MDA-MB-231 and MCF10CA1a breast cancer cell lines when they first introduce them in line 80. Why did they choose these two cell lines for their experiments?

Authors' response:

We have added sentences that describe the differences between the MDA-MB-231 and MCF10CA1a cell line after line 80 in the manuscript, as recommended by the reviewer. In our previous work (ref.17 in the manuscript), we discovered and reported profound biological effects of iEFs on the migratory properties and actin cytoskeleton in SCP2 cells (which are a single cell-derived progeny obtained from parental MDA-MB-231 cells and which are highly metastatic clones of these parental cells). Our earlier work and the relationship of those cell lines (SCP2) to the parental MDA-MB-231 cell line dictated the choice of MDA-MB-231 cells for the work reported in this manuscript.

To benchmark the effects of iEFs on cancer cells, we also used MCF10A cells which are a model for normal mammary epithelial cells. Moreover, to understand if iEFs selectively alter motility responses of metastatic cells we used the MCF10CA1a cell lines which were generated by inducing selective mutations (H-Ras mutation in culture followed by *in vivo* selection in mice) in MCF10A cells. These mutations transformed the non-metastatic MCF10A cells into highly malignant and metastatic MCF10CA1a cells (ref 18, 19 and 20 in the revised manuscript). This helped us get a better understanding of the effects of iEFs on tumorigenic versus non-tumorigenic cell lines. Conducting experiments with the MCF10CA1a helped establish that the effects of iEFs are cell line dependent and strongly suggest that iEFs selectively alter the motility of metastatic breast cancer cells.

Reviewer's comment #2:

The supplementary figures should be presented in the order of their first mention in the results section, e.g. Fig. S3 is mentioned first in line 108, followed by Fig S2 in line 110 and Fig. S1 in line 121. Therefore, I recommend renaming Fig S3 to S1 and Fig S1 to S3 and showing them in the correct order in the supplementary material.

Authors' response:

The reviewer is correct that the figures in the supplementary information section are not numbered in the order they are referenced. We apologize for this oversight and have corrected it in the revised manuscript. However, we do wish to point out to the reviewer that the first time Fig S1 is referenced is in the first line of Results with Fig 1 A-C, S1, in the main article. This has been changed in main manuscript text to Fig 1 A-C, Fig S1 to avoid any confusion. The supplementary figures are now numbered in order from the first time they are referenced in the main article.

Reviewer's comment #3:

In most of their figures the authors compare the three cell lines MDA-MB-231, MCF10CA1a and MCF10CA, except in Figures 2 and 7. I recommend also showing the “normal” cell line MCF10CA in these figures for consistency, even though they did not migrate as explained in lines 148-155 of the results section - or at least to comment briefly in the figure legend on why they are not shown to avoid confusion of the reader when looking at the figure without having read those lines already.

Authors' response:

We presume that the review is referring to the “normal” cell line as MCF10A and not MCF10CA. Figure 2 shows data for cell migration in the *absence* of EGF gradients. In the absence of exogenous EGF gradients, the “normal” cells MCF10A do not display a migratory phenotype and hence these are not shown in Fig. 2. The same is true for Fig. 7. We have added a statement in the captions of Figure 2 and Figure 7 clearly stating that the “normal” MCF10A cells do not migrate in these conditions to clear any confusion for the readers.

Reviewer's comment #4:

The x-axis is crossing the y-axis at different positions for the different cell lines but at similar positions in various experiments. I assume that this refers to the migration speed and persistence

of the respective untreated control cells from Figure 2 as a reference? However, this needs to be explained in the figure legends for a better understanding.

Authors' response:

We believe the reviewer is referring to the horizontal lines indicating the mean speeds and persistence of the different cell lines in Figs. 2, 3, and 7, when referring to the “x-axis crossing the y-axis at different positions for the different cell lines...” These lines indicate the mean values for the baseline or control case of no iEF, no EGF, and/or no Akt inhibition. Each of the cell types that we studied has a different value for mean speeds and persistence for the baseline or control case, which explains why the horizontal lines intersect the y-axis at different locations for the different cell types. **In response to the reviewer comment, we have added a sentence clarifying this point in each of these figures in the revised manuscript.** We note that the desired effect of these horizontal lines in each of the graphs was to provide immediate visual confirmation for the readers of the stimulatory or inhibitory effect of the experimental conditions relative to the baseline or control condition. We believe that the added clarifying sentences will help ensure this outcome.

Reviewer's comment #5:

In the discussion lines 437, 439 and 454 two tables are mentioned, but there are no tables provided within the manuscript or supplementary files

Authors' response:

We thank the reviewer for catching this oversight in our manuscript. **Our revised manuscript addresses this deficiency.**

Reviewer's comment #6:

In lines 524 and 526 the authors refer to TTFs as “electromagnetic therapy”. TTFs are electric fields of low intensity and intermediate frequency. They are not electromagnetic fields.

Authors' response:

Based on our understanding of the literature on TTFs, these are time-varying electric fields (E). According to Maxwell's equations, time varying electric fields can produce currents that in turn develop time-varying magnetic fields (B). These time-varying magnetic fields will in turn modify the exogenously applied time-varying electric fields as stated by the Faraday's Law below. We therefore believe we are justified in using the term “electromagnetic therapy” to include TTFs.

$$\text{Faraday's Law: } \nabla \times E = - \frac{\partial B}{\partial t}$$

Therefore, situations with either time-varying electric fields or time-varying magnetic field can both be referred to broadly as electromagnetic in nature.

Reviewer's comment #7:

The authors provide a supplementary video, but they do not refer to this video in their manuscript.

Authors' response:

We thank the reviewer for pointing out this oversight as well. This has been corrected in the revised manuscript. We have referenced the video in the main manuscript to help put it in context of our results and findings.

Reviewer's comment #8:

In the materials and methods section there seems to be some mix-up in cross-referencing to the supplementary figures. In lines 612-614 the authors speak about the measurements of the Helmholtz coil and refer to figure S6. However, Figure S6 shows the effect of iEFs on levels of p-Akt. Probably the authors mean Figure S7? A similar error seems to be in lines 684, 687 and 792/793 when referring to (the wrong) Figures S5 and S6. This requires correction.

Authors' response:

We thank the author for catching these errors in our manuscript, which have been corrected in the revised manuscript.

Reviewer's comment #9:

This reviewer is working on Tumor Treating Fields (TTFields). TTFields application is inducing heat in the culture dish which needs to be tightly controlled. Do iEFs induce any temperature change within the microfluidic bi-directional microtrack assay and how is this controlled?

Authors' response:

The reviewer is very astute in bringing up the point of heating, which is a key distinguishing difference between TTFs and iEFs. In TTFs, the electric fields are on the order of 1 V/cm *and higher*, whereas the iEFs in our work are on the order of 100 $\mu\text{V}/\text{cm}$ *or less*. Since the Ohmic dissipation scales as σE^2 , where σ is the electrical conductivity of the medium containing the cells and E is the electric field, for comparable volumes and electrical properties of the media used, the volumetric heating with iEFs is 10^8 times smaller than with TTFs. Hence, heating is not an issue in our experiments and temperature is easily maintained within the microchannels by the constant environment provided by the incubator and associated conductive and convective heat transfer.

Reviewer's comment #10:

In the legend of figure 4, line 938 it should read "iEFs have no effect on EGFR distribution..."

Authors' response:

We believe the reviewer may have misread the caption. Line 938 in the caption of Fig. 4 does indeed state "iEFs have no effect on EGFR distribution ..."

Reviewer's comment #11:

In the legend of Figure S6, line 2 the word "levels" appears twice.

Authors' response:

We thank the author for catching this error in our manuscript, which has been corrected in the revised manuscript.

Reviewer's comment #12:

The authors should decide whether they like to put a space character between numerical value and unit or not. This is jumbled throughout the manuscript.

Authors' response:

We appreciate this point raised by the reviewer. We have applied a consistent standard now throughout our revised manuscript. All numerical values are now given as: 0.33 $\mu\text{m}/\text{min}$. We added a space between the numerical value and the units that follow.

We thank the reviewer for the comments and questions raised in the review as we believe it significantly enhances the readability of our article and makes apparent the significant contributions of our work. We have made revisions to our original manuscript taking into account all of this reviewer's helpful comments and criticisms, and further addressed the reviewer's concerns and questions in this response. We believe our revised manuscript is now worthy of publication in *Communications Biology*.

REVIEWERS' COMMENTS:

Reviewer #1 (Remarks to the Author):

The authors have thoroughly and adequately addressed the concerns raised by the reviewers. This reviewer recommends acceptance for publication.

Reviewer #2 (Remarks to the Author):

The authors Garg et al. provided a revised version of their manuscript "Electromagnetic fields alter the motility of metastatic breast cancer cells". They convincingly addressed all my concerns and questions in their revised manuscript and rebuttal. I recognized just two minor errors: On page 26, line 19 it should read "our" instead of "out" and the three panels of Figures 4 and S5 are labeled "A", "C" and "E", but referred to in figure legend and text as "A", "B" and "C". However, these are very minor corrections and I now support publication of this highly interesting work in Communications Biology.

Reviewer #2 (Remarks to the Author):

The authors Garg et al. provided a revised version of their manuscript "Electromagnetic fields alter the motility of metastatic breast cancer cells". They convincingly addressed all my concerns and questions in their revised manuscript and rebuttal. I recognized just two minor errors: On page 26, line 19 it should read "our" instead of "out" and the three panels of Figures 4 and S5 are labeled "A", "C" and "E", but referred to in figure legend and text as "A", "B" and "C". However, these are very minor corrections and I now support publication of this highly interesting work in Communications Biology.

Author response: These errors in Figures 4 and S5 (labels "A", "C", and "E") have been corrected to "A", "B", and "C" in the latest revision.